# Chronic hyperactivation of midbrain dopamine neurons causes preferential dopamine neuron degeneration

Katerina Rademacher[1,2,3†], Zak Doric[1,2†], Dominik Haddad[1†], Aphroditi Mamaligas[1], Szu-Chi Liao[1,3,4,5], Rose Creed[3,6], Kohei Kano[1,3], Zac Chatterton[3,7], Yuhong Fu[3,7], Joseph H Garcia[1,8], Victoria M Vance[1,3,9], Yoshitaka J Sei[1,3], Anatol Kreitzer[1,2,10], Glenda Halliday[3,7], Alexandra B Nelson[2,3,6], Elyssa Margolis[2,6], Ken Nakamura[1,2,3,6,11]*

[1]Gladstone Institute for Neurological Disease, Gladstone Institutes, San Francisco, United States; [2]Graduate Program in Neuroscience, University of California San Francisco, San Francisco, United States; [3]Aligning Science Across Parkinson's (ASAP) Collaborative Research Network, Chevy Chase, United States; [4]Department of Nutritional Sciences & Toxicology, University of California Berkeley, Berkeley, United States; [5]Endocrinology Graduate Program, University of California Berkeley, Berkeley, United States; [6]UCSF Weill Institute for Neurosciences, Department of Neurology, University of California San Francisco, San Francisco, United States; [7]Brain and Mind Centre, Faculty of Medicine and Health, School of Medical Sciences, University of Sydney, Sydney, Australia; [8]School of Medicine, University of California San Francisco, San Francisco, United States; [9]College of Science, Northeastern University, Boston, United States; [10]UCSF Department of Physiology, University of California San Francisco, San Francisco, United States; [11]Graduate Program in Biomedical Sciences, University of California San Francisco, San Francisco, United States

*For correspondence:
ken.nakamura@gladstone.ucsf.edu

†These authors contributed equally to this work

Competing interest: The authors declare that no competing interests exist.

## eLife Assessment

This manuscript shows that chronic chemogenetic excitation of dopaminergic neurons in the mouse midbrain results in differential degeneration of axons and somas across distinct regions (SNc vs VTA). These findings are **important** for two reasons. This approach can be used as a mouse model for Parkinson's Disease without the need for the infusion of toxins (e.g. 6-OHDA or MPTP) — this mouse model also has the advantage of showing axon-first degeneration over a time course (2–4 weeks) that is suitable for experimental investigation. Also, the findings that direct excitation of dopaminergic neurons causes differential degeneration sheds light on the mechanisms of dopaminergic neuron selective vulnerability. The evidence that activation of dopaminergic neurons causes degeneration, alters motor behavior, and alters mRNA expression is **convincing**. This is an exciting paper that will have an impact on the Parkinson's Disease field.

**Abstract** Parkinson's disease (PD) is characterized by the death of substantia nigra pars compacta (SNc) dopamine (DA) neurons, but the pathophysiological mechanisms that precede and drive their death remain unknown. The activity of DA neurons is likely altered in PD, but we understand little about if or how chronic changes in activity may contribute to degeneration. To address this question, we developed a chemogenetic (DREADD) mouse model to chronically increase DA neuron activity and confirmed this increase using ex vivo electrophysiology. Chronic hyperactivation

of DA neurons resulted in prolonged increases in locomotor activity during the light cycle and decreases during the dark cycle, consistent with chronic changes in DA release and circadian disturbances. We also observed early, preferential degeneration of SNc projections, recapitulating the PD hallmarks of selective vulnerability of SNc axons and the comparative resilience of ventral tegmental area axons. This was followed by the eventual loss of midbrain DA neurons. Continuous DREADD activation resulted in a sustained increase in baseline calcium levels, supporting a role for increased calcium in the neurodegeneration process. Finally, spatial transcriptomics from DREADD mice examining midbrain DA neurons and striatal targets, and cross-validation with human patient samples, provided insights into potential mechanisms of hyperactivity-induced toxicity and PD. Our results thus reveal the preferential vulnerability of SNc DA neurons to increased neural activity and support a potential role for increased neural activity in driving degeneration in PD.

## Introduction

In Parkinson's disease (PD), the loss of substantia nigra pars compacta (SNc) dopamine (DA) neurons leads to severe disruption of circuit dynamics in the basal ganglia. Compensation for DA loss involves changes in the activity of both surviving SNc neurons and of other downstream neurons in the circuit. Indeed, following lesions of the nigrostriatal pathway in rats, surviving SNc DA neurons are hyperactive (*Hollerman and Grace, 1990*), release additional DA (*Zigmond et al., 1984*; *Agid et al., 1973*; *Hefti et al., 1980*; *Zhang et al., 1988*), and have reduced DA reuptake (*Zigmond et al., 1984*). Massive loss of DA neurons (*Hollerman and Grace, 1990*; *Chen et al., 2009*; *Stachowiak et al., 1987*), complete loss of mitochondrial complex I activity (*González-Rodríguez et al., 2021*), and loss of the mitochondrial PD protein PINK1 (*Bishop et al., 2010*) can also result in increased burst firing (*Schiemann et al., 2012*; *Morikawa and Paladini, 2011*). Therefore, DA neurons are predisposed to altered activity in the setting of extensive loss or stress, which may drive ongoing disease processes. Some lines of evidence support the potential role of hyperactivity in disease initiation, including increased activity in a subset of DA neurons before their degeneration in MitoPark mice (*Good et al., 2011*), increased spontaneous firing in PD patient-derived iPSC DA neurons (*Lin et al., 2021*), and increased activity of nigrostriatal DA neurons in genetic models of PD (*Bishop et al., 2010*; *Regoni et al., 2020*). Moreover, critical PD disease proteins including α-synuclein, LRRK2, PINK1, and Parkin can influence the level of neural activity (*Regoni et al., 2020*; *Dagra et al., 2021*; *Janezic et al., 2013*; *Dodson et al., 2016*; *Subramaniam et al., 2014*; *Chou et al., 2019*). In particular, the normal function of α-synuclein is believed to be regulating neural activity (*Nemani et al., 2010*), further supporting the idea that changes in neural activity may contribute to disease pathophysiology. Circuit-level changes may also contribute to adverse DA neuron activity. For instance, evidence from primate models suggests that the subthalamic nucleus, which sends a glutamatergic projection to the SNc, is hyperactive in PD (*Bergman et al., 1994*). While system-level changes may be compensatory and partially restore DA levels and overall motor function, they may also have adverse consequences.

Healthy SNc DA neurons are believed to have immense energetic requirements due to their pacemaking activity, active calcium pumping, unmyelinated, or poorly myelinated fibers (*Braak and Del Tredici, 2004*; *Tagliaferro and Burke, 2016*), and large axonal arbors (*Haddad and Nakamura, 2015*). This large energetic requirement likely accounts for their intrinsic vulnerability to mitochondrial insults, including complex I disruption (*González-Rodríguez et al., 2021*; *Betarbet et al., 2000*; *Doric and Nakamura, 2021*) and impairments in mitochondrial dynamics (*Berthet et al., 2014*) and turnover (*Li et al., 2021*). Neighboring ventral tegmental area (VTA) neurons are relatively spared in PD (*German et al., 1992*; *Kish et al., 1988*), and this may be due to lower reliance on calcium oscillations for pacemaking (*Khaliq and Bean, 2010*), their ability to buffer calcium more effectively than SNc neurons (*German et al., 1992*), and smaller axon arbors (*Matsuda et al., 2009*; *Surmeier and Schumacker, 2013*; *Pacelli et al., 2015*; *Giguère et al., 2019*). Thus, combined with disease-related stress, the metabolic impact of even mild hyperactivity may trigger or accelerate degeneration in SNc DA neurons. In support of this hypothesis, inhibiting the excitatory input from the STN protects SNc DA neurons from 6-OHDA and MPTP toxicity (*Piallat et al., 1996*; *Bergman et al., 1990*). However, empirical evidence linking chronic changes in neural activity to the degeneration of SNc DA neurons in PD is lacking. Recordings from putative SNc neurons in PD patients displayed twofold higher burst firing when compared to recordings from healthy rodents and nonhuman primates despite similar

mean firing rates, though this data is difficult to interpret without human controls (*Schiemann et al., 2012*; *Zaghloul et al., 2009*).

Additionally, changes in calcium dynamics during hyperactivity can also drive metabolic stress. SNc DA neurons rely on voltage-gated $Ca_v1.3$ calcium channels to support pacemaking, and blocking these channels is protective against the toxicity of 6-OHDA and MPTP (*Wang et al., 2017*; *Ilijic et al., 2011*). While classic excitotoxicity involves cytosolic calcium overload and acute cell death, chronic synaptic hyperactivity results in sublethal stress to mitochondria, promoting calcium dysregulation and dendritic atrophy (*Verma et al., 2022*). Mitochondrial calcium overload has been observed in the setting of *Lrrk2* mutation and PINK1 deficiency (*Verma et al., 2022*). Therefore, altered activity, metabolic stress, and calcium overload may all contribute to DA neuron death.

To understand if chronic hyperactivation of DA neurons is sufficient to cause neurodegeneration, we developed a chemogenetic mouse model. Our results indicate that chronically increasing neural activity in midbrain DA neurons results in alteration of circadian locomotion patterns, and prolonged activation leads to selective degeneration of SNc axons and eventual death of midbrain DA neurons. These changes were accompanied by altered intracellular calcium dynamics and transcriptomic changes consistent with calcium dysregulation, supporting a role for increased neural activity in driving neurodegeneration in PD.

## Results

To model a chronic increase in DA neuron activity, as may occur in PD, we used a chemogenetic approach. We first expressed the excitatory DREADD hM3Dq specifically in DA neurons using stereotaxic delivery of Cre-dependent hM3Dq AAV to the SNc and VTA of mice expressing Cre under the DA transporter promoter (*Slc6a3^IRES-Cre^* or DAT^IREScre^). Next, we measured the acute behavioral effects of chemogenetic activation of DA neurons. As locomotor output is strongly tied to nigrostriatal DA function (*da Silva et al., 2018*; *Jin and Costa, 2010*; *Kravitz et al., 2010*), we used home cage wheel running as an in vivo proxy for changes in DA function. Mice were single-housed, and locomotion was quantified based on wheel rotation. Two weeks after viral injection, we administered clozapine-*N*-oxide (CNO) by i.p. injection (0.5 mg/kg) and confirmed that mice responded with an acute increase in wheel running as an indicator of successful DREADD expression (*Figure 1—figure supplement 1A*). The resulting hM3Dq-expressing DAT^IREScre^ mice were then randomly assigned to be administered either vehicle (2% sucrose, to compensate for CNO's bitter flavor [*Kumar et al., 2024*]) or CNO (2% sucrose, 300 mg/L CNO) via drinking water ad libitum for 2 weeks (*Figure 1A*). This strategy allowed us to chronically activate SNc and VTA DA neurons.

Previous reports indicate that chemogenetic activation of DA neurons leads to increased locomotion (*Wang et al., 2013*), but the effects of chronic, long-term activation remain unknown. During the first day of treatment, CNO-treated animals were more active than controls during the dark cycle and also strongly trended toward more activity during the light cycle (*Figure 1B and C*). By day 3 of treatment, dark cycle activity markedly decreased in CNO-treated DREADD-expressing mice and remained decreased for the rest of the treatment time, whereas activity during the light cycle remained increased (*Figure 1B and C*, *Figure 1—figure supplement 1B*). Decreased wheel usage in CNO-treated animals may reflect a consequence of circadian disruption. The prolonged changes in wheel activity indicate that the behavioral effects of chemogenetic activation persist throughout the 2-week testing period. In contrast, when DAT^IREScre^ mice were given CNO for 4 weeks in the absence of the DREADD (CNO alone), there were no significant changes in activity in either light or dark cycles (*Figure 1—figure supplement 1C*). These results indicate that CNO alone does not significantly alter mouse activity.

We performed a similar experiment administering CNO for 2 weeks to DAT^IREScre^ mice injected with hM3Dq that did not respond to acute CNO IP injection (nonresponders; *Figure 1—figure supplement 1D*). Nonresponders did not show strong initial changes to light or dark cycle wheel usage in the first few days of treatment but did show persistent decreases in dark cycle usage. Presumably, the lack of early changes reflects either differences in the amount of hM3Dq expressed or the proportion of neurons that are transfected.

To gain insight into the impact of prolonged chemogenetic activation on DA neuron electrophysiology, we performed ex vivo whole-cell recordings in DREADD-expressing SNc neurons in midbrain slices after 7 days of in vivo treatment with CNO or vehicle. In vivo DREADD activation induced several

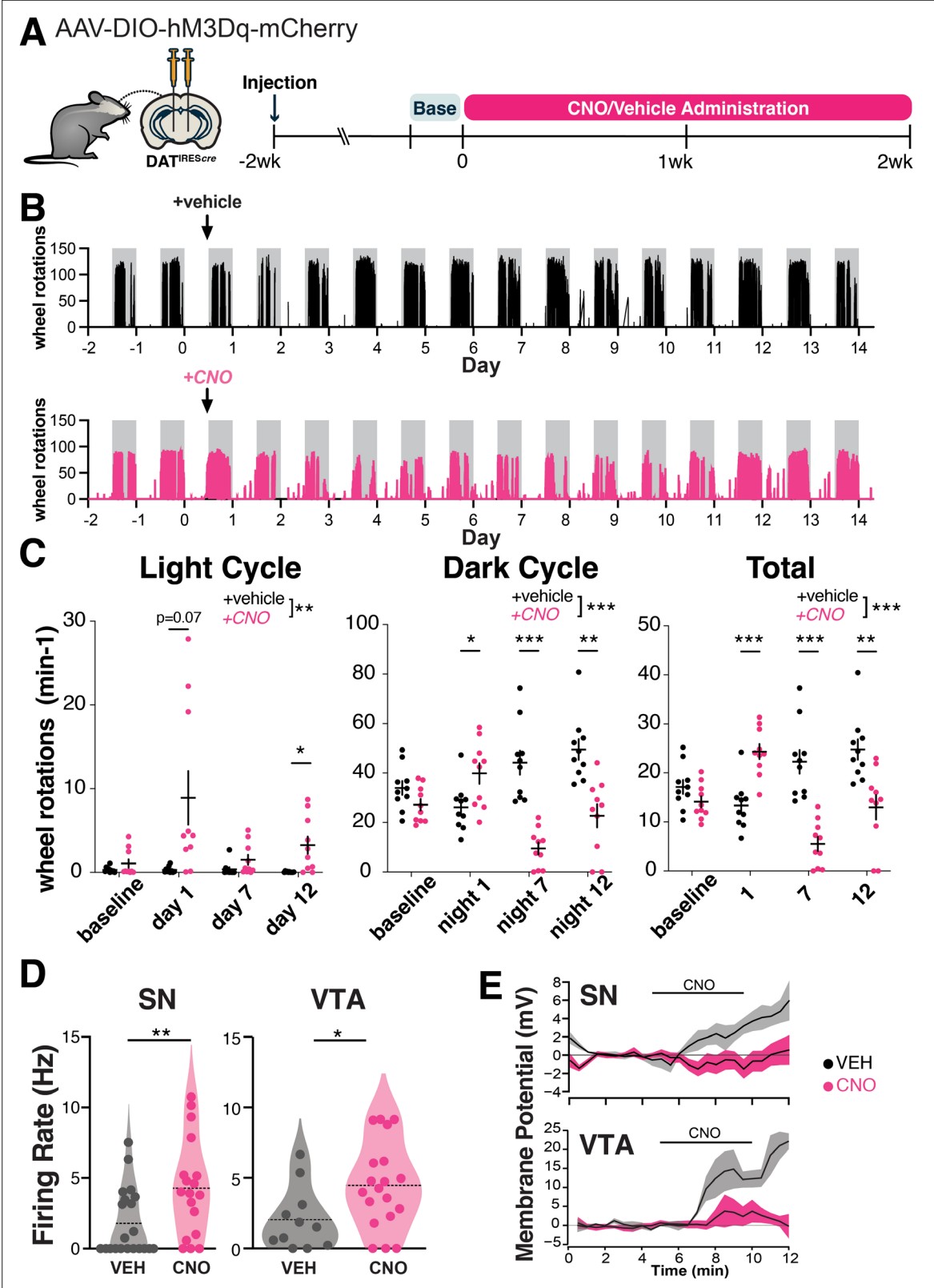

**Figure 1.** Chronic hM3Dq activation persistently alters the activity of substantia nigra pars compacta (SNc) dopamine neurons. (**A**) Graphical illustration summarizing experimental design. Recombinant AAV encoding a conditional allele of the hM3Dq(DREADD)-mCherry was injected bilaterally into the ventral midbrain of 4- to 5-month-old DAT*IREScre* mice. CNO (300 mg/L) or vehicle (2% sucrose in water) was administered ad libitum via drinking water for 2 weeks, and the animals were perfused the next day. Changes in locomotion were assessed with running wheels. The 2 days preceding the start

*Figure 1 continued on next page*

*Figure 1 continued*

of treatment were used as a measure for baseline locomotion. (**B**) Representative traces of wheel usage for animals given control vehicle water (top) or CNO water (bottom). Arrows denote start of treatment. Gray background shading indicates dark cycle hours. (**C**) Mean wheel usage for selected days during the experiment, segregated by light (left), dark (middle), and total (light+dark; right) cycles. n=10 animals/group from 2 independent experiments. (**D**) Spontaneous firing rate was measured during the first 2 min of whole-cell recordings. n = 21 neurons from 4 vehicle-treated mice and 18 neurons from 5 CNO-treated mice for SN and 11 neurons from 2 vehicle-treated mice and 19 neurons from 2 CNO-treated mice for VTA. (**E**) Time course of responses to bath application of 1 μM CNO ex vivo measured in current clamp in neurons from vehicle-treated and CNO-treated mice. n = 5 neurons from vehicle-treated mice and 9 neurons from CNO-treated mice for SN and 3 neurons from vehicle-treated mice and 7 neurons from CNO-treated mice. Data indicate mean ± SEM. *p≤0.05, **p<0.01, ***p<0.001 by two-way ANOVA and Holm-Sidak post hoc test (**C**). *p≤0.05, **p=0.01 by t-test or permutation (nonparametric) analysis (**D**).

The online version of this article includes the following figure supplement(s) for figure 1:

**Figure supplement 1.** Additional behavior data.

**Figure supplement 2.** Additional electrophysiology data.

changes in somatodendritic properties of SNc neurons, including a marked decrease in the magnitude of the hyperpolarization-activated nonselective cation current $I_h$ (***Figure 1—figure supplement 2A***). It also led to an increase in the spontaneous firing rate (***Figure 1D***, ***Figure 1—figure supplement 2C***), indicating a persistent increase in firing rate with CNO treatment. The rate of firing was decreased somewhat in controls relative to historical controls from our lab (***Berthet et al., 2014***), possibly reflecting mild cellular stress from the AAV virus. We did not observe a difference in the coefficient of variation of the interspike interval between treatment groups; thus, there was no change in the regularity of pacemaker firing (***Figure 1—figure supplement 2A***). In vivo CNO led to a more depolarized action potential (AP) peak voltage in spontaneous APs (***Figure 1—figure supplement 2A***). CNO treatment did not impact AP threshold voltage or duration (***Figure 1—figure supplement 2A***), indicating that AP waveforms were not degraded.

To measure the direct physiological impact of DREADD activation in SNc neurons, we also acutely bath-applied 1 μM CNO to these slices. Interestingly, while SNc neurons from CNO-naïve hM3Dq mice depolarized in response to acute CNO, chronic exposure to CNO for 1 week in vivo eliminated this acute response (in neurons from vehicle-treated mice: 4.9±2.9 mV [n=5], in neurons from CNO-treated mice: –0.5±1.0 mV [n=9]; p=0.05, t-test, ***Figure 1E***, ***Figure 1—figure supplement 2D***). Taken together, these findings indicate that 7 days of CNO treatment increased spontaneous firing but altered hM3Dq function such that acute physiological impacts were no longer apparent. The absence of an acute response may indicate a change in the coupling or availability of the receptors, an adaptation or dysfunction within the neurons, or a homeostatic response to prolonged activation. This may also reflect early stages of toxicity to the CNO-treated DA neurons.

We performed similar recordings in DREADD-expressing VTA neurons. As in the SNc, we found that the spontaneous firing rate was higher with 7 days of in vivo DREADD activation (***Figure 1D***, ***Figure 1—figure supplement 2C***). Unlike the SNc, the coefficient of variation of the interspike interval was decreased in VTA neurons from the CNO-treated mice, indicating more regular firing patterns (***Figure 1—figure supplement 2B***). Also unlike the SNc, VTA neurons did not display changes in $I_h$ magnitude or changes in AP peak voltage in spontaneous APs following chronic in vivo DREADD activation (***Figure 1—figure supplement 2B***). In vivo treatment did not impact AP threshold voltage, AP peak, AP duration, input resistance, or initial membrane potential in VTA neurons (***Figure 1—figure supplement 2B***).

In response to acute bath application of CNO to slices, VTA neurons from CNO-naïve hM3Dq mice depolarized, while those from mice that received chronic 7-day exposure in vivo to CNO showed markedly reduced but not fully eliminated responses (in neurons from vehicle-treated mice: 11.2±8.5 mV [n=3], in neurons from CNO-treated mice: 2.0±2.2 mV [n=7]; p=0.17 t-test, ***Figure 1E***, ***Figure 1—figure supplement 2D***), indicating some retained hM3Dq functionality. Together, these data may reflect resilience to chronic activation in VTA neurons compared to SNc neurons.

## SNc axons are preferentially vulnerable to chronic hM3Dq activation

To determine if chronic chemogenetic activation of DA neurons for 2 weeks induces degeneration, we first quantified its effects on axonal integrity in the striatum. Strikingly, compared to vehicle-treated mice, hM3Dq-expressing animals treated with CNO lost ≈40% of their dopaminergic axons in the

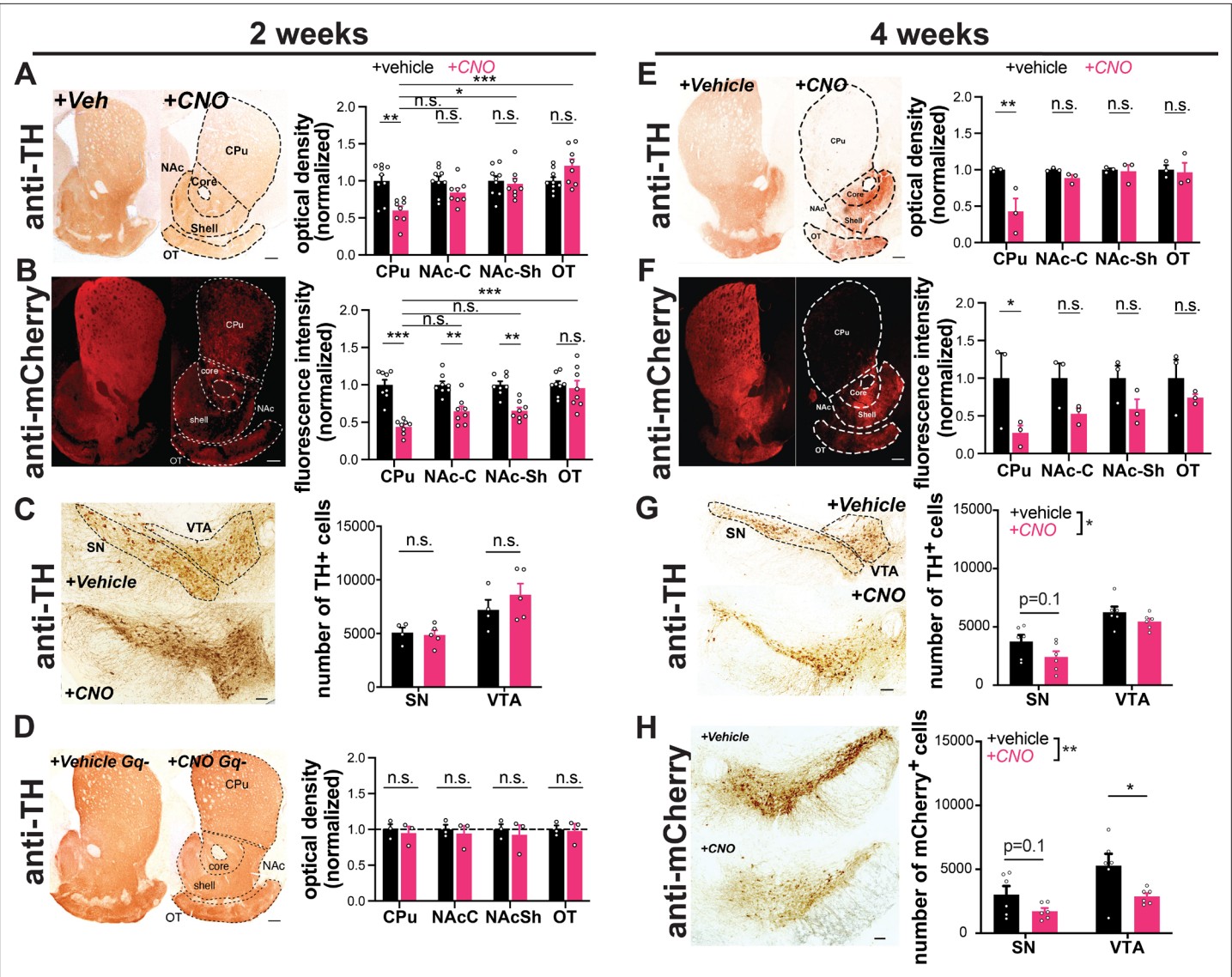

**Figure 2.** Chronic activation with AAV-hM3Dq-DREADDs is preferentially toxic to nigrostriatal axons. DAT$^{IREScre}$ mice expressing hM3Dq(DREADD)-mCherry (**A–C, E–H**) or no virus injection (**D**) in dopamine (DA) neurons. Example images of TH (**A,D,E**) and mCherry (**B,F**) immunoreactivity in striatal sections of mice treated for 2 or 4 weeks with vehicle (left) or clozapine-*N*-oxide (CNO) (right) via drinking water. DA neuron projection areas in dorsal and ventral striatum are indicated with dotted lines. Quantifications for TH (**A,D,E**) and mCherry (**B,F**) optical density at 2 and 4 weeks show preferential loss in CPu. n=8–9 (**A**), 8 (**B**), or 3 (**D, E, F**) animals/group, 3–5 sections/animal. (**C,G,H**) Images of TH and mCherry (4 weeks only) immunoreactivity in midbrain of vehicle (top) or CNO (bottom) treated mice at 2 or 4 weeks. SN and VTA regions are indicated with dotted lines. (**C,G,H**) Stereology estimating the number of TH+ or mCherry+ DA neurons. Chronic CNO treatment of hM3Dq(DREADD)-expressing mice shows a significant decrease in both TH and mCherry immunoreactivity and DA neuron number. n=4–5 hemispheres (one per mouse) (**C**) or 6 hemispheres (two per mouse) (**G, H**) per group. Scale bars indicate 100 μm in the midbrain and 200 μm in the striatum. Data indicate mean ± SEM. *p<0.05, **p<0.01, ***p<0.001 by two-way ANOVA and Holm-Sidak post hoc test. n.s.: not significant. SN: substantia nigra, VTA: ventral tegmental area, CPu: caudate putamen, NAc-C: nucleus accumbens core, NAc-Sh: nucleus accumbens shell, OT: olfactory tubercle.

The online version of this article includes the following figure supplement(s) for figure 2:

**Figure supplement 1.** Additional histological data.

dorsal striatum, as evidenced by decreases in both TH immunoreactivity and reporter mCherry immunofluorescence (*Figure 2A and B*). The same perturbation had a lesser impact on nucleus accumbens and no impact on olfactory tubercle DA afferents. Notably, TH immunoreactivity and mCherry immunofluorescence also decreased in nonresponder mice following 2 weeks of treatment (*Figure 2—figure supplement 1A*), although the extent of decrease was somewhat less than in responders.

After a shorter 1-week treatment period in a small cohort, we also observed a trend for decreased TH immunoreactivity in the CPu (*Figure 2—figure supplement 1B*).

We also assessed the impact of 2 weeks of activation on DA neuron survival in the midbrain. With stereological quantification, we found no difference in neuron number between CNO-treated animals and vehicle-treated controls in the SNc or VTA (*Figure 2C*). Therefore, at this time point, CNO-treated mice exhibit severe axonal degeneration without DA neuron death.

To ensure that the loss of DA terminals was not due to an off-target effect of CNO, we assessed TH expression in CNO alone animals. These mice did not show a decrease in dopaminergic axons in the CP (*Figure 2D*), supporting the conclusion that the axonal degeneration requires hM3Dq activation.

We next assessed the impact of more prolonged activation on degeneration of DA neuron somata. Mice treated with CNO for 4 weeks lost the majority of axons projecting to the CP (*Figure 2E and F*), while fibers in the nucleus accumbens and olfactory tubercle were again preferentially spared. Moreover, following 4 weeks of CNO treatment, there was a significant decrease in the overall number of TH+ midbrain (SNc and VTA) DA neurons, assessed by stereology (*Figure 2G and H*). Although the loss of TH reactivity can occur in the absence of neuronal death, there was a similar decrease in the number of mCherry+ DA neurons (labeled by DsRed) in the midbrain. The alignment of these findings supports true loss of DA neurons, although quantitation of mCherry+ DA neurons specifically informs the impact on transduced DA neurons, and this analysis does not account for the variability between injections. Together, these data suggest that, similar to PD progression in humans (*Kordower et al., 2013*), axonal impairment precedes cell body loss in this model. No preferential loss of SNc vs VTA cell bodies was detected at this time point.

## Chronic activation increases baseline population calcium levels in midbrain DA neurons

hM3Dq activation may increase neural activity by increasing intracellular calcium levels (*Roth, 2016*), and increased neural activity should also increase intracellular calcium. To determine the impact of 2 weeks of hM3Dq activation on intracellular calcium concentrations in DA neurons, we bred DAT$^{IREScre}$ mice to Ai148D mice that contain a Cre-dependent GCaMP6f calcium indicator and injected them bilaterally with the Cre-dependent hM3Dq AAV (*Figure 3A*). The fluorescent signal was used to guide implantation of an optical fiber in the midbrain of animals (*Figure 3B and C*), with confirmed hM3Dq expression based on increased wheel running in response to i.p. CNO (*Figure 1—figure supplement 1A*). Mice were administered CNO or vehicle water and then placed in arenas for 10 min fiber photometry recording sessions every 1–3 days over the 2-week treatment period. Mean fluorescence levels served as a proxy for population-level intracellular calcium concentrations. We observed a small trend for increased fluorescent signal the day after starting CNO that persisted for days 3 through 5 but did not reach significance (*Figure 3D–F*). Interestingly, this was followed by a second, much larger increase in calcium levels between 10 and 17 days (*Figure 3D–F*), that occurred in parallel with axon loss (*Figure 2A*, *Figure 2—figure supplement 1B*). To determine if the increase in neural activity was reversible, we performed two additional recording sessions after removing CNO from the drinking water. Although there was a small trend for decreased fluorescence, this did not reach significance.

Although GCaMP fluorescence is not quantitative, the consistent difference in baseline in animals tested concurrently on the same system provides confidence in our interpretation. The frequency and amplitude of calcium transients decreased with chronic CNO treatment, and during washout, the frequency remained decreased while the amplitude returned to vehicle levels (*Figure 3—figure supplement 1A*). Notably, the decrease in frequency or amplitude is unlikely to represent a ceiling effect (i.e. where a high baseline might occlude transient detection), because while the absolute baseline and transient amplitude returned to pre-treatment levels during the washout, the frequency of transients did not. Instead, the decreases in transient frequency and amplitude may reflect less bursting activity in the neurons. As sharp peaks in genetically encoded calcium indicator signals tend to reflect bursting activity, the sustained decrease in frequency during washout may reflect more permanent dysfunction in the burst firing patterns of these neurons.

Together, these data support massive increases in DA neuron intracellular calcium over time, which may be due to hM3Dq receptor activation, calcium dysregulation, and the onset of degeneration. We also assessed time spent moving and distance traveled during photometry sessions in a subset of animals (*Figure 3—figure supplement 1B*). Paralleling the small initial increase in baseline calcium,

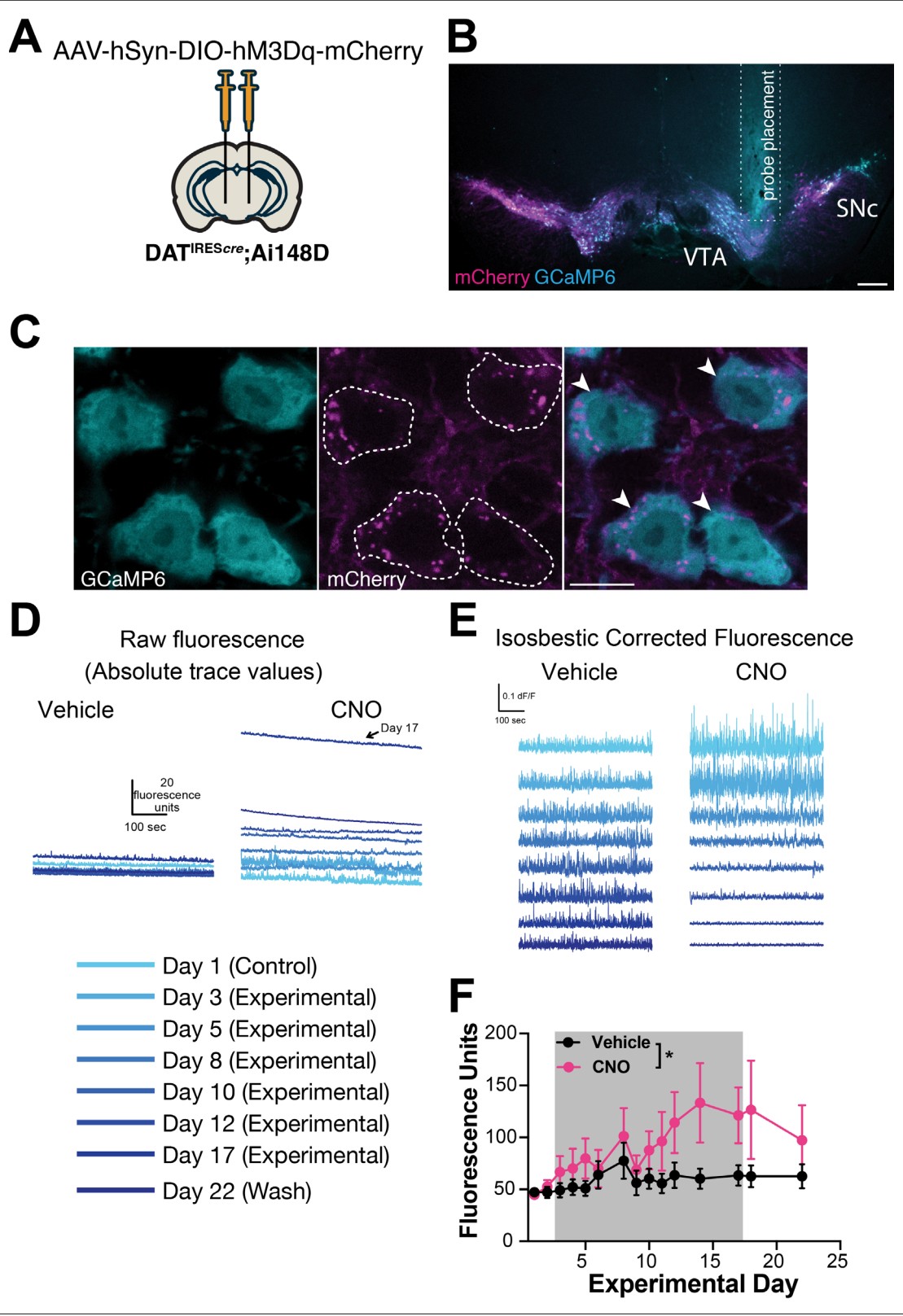

**Figure 3.** Chronic hM3Dq activation increases baseline calcium in parallel with axonal degeneration. (**A**) Transgenic Ai148D mice expressing a Cre-dependent calcium reporter GCaMP6f were crossed with DAT*IREScre* to express GCaMP6f specifically in dopamine (DA) neurons. Mice were injected bilaterally with AAV-DIO-hM3Dq-mCherry and implanted with an optical probe for baseline calcium measurements during a 14-day chronic chemogenetic activation. (**B**) Representative image of photometry probe placement in mouse midbrain to record from DA neurons co-expressing

*Figure 3 continued on next page*

Figure 3 continued

hM3Dq-mCherry (magenta) and GCaMP6 (cyan). Scale bar is 200 µm. (**C**) Representative high-magnification images of reporter mCherry (magenta) and GCaMP6 (cyan) co-expression. Scale bar is 10 µm. (**D**) Representative raw traces of baseline calcium fluorescence in mice treated with vehicle vs clozapine-*N*-oxide (CNO). (**E**) Representative isosbestic corrected traces in mice treated with vehicle vs CNO. (**F**) Baseline calcium fluorescence levels of DA neurons in mice treated with vehicle or CNO for 14 days (gray shaded area) and following wash. n=7 mice/group from 2 independent experiments. Data indicate mean ± SEM. *p<0.05 by two-way ANOVA followed by Holm-Sidak post hoc test.

The online version of this article includes the following figure supplement(s) for figure 3:

**Figure supplement 1.** Additional photometry data.

there was a strong trend for CNO to increase activity over time vs controls in both measures of gross locomotion during the first week of treatment (two-way ANOVA distance traveled p=0.07, percent time spent moving p=0.06). Interestingly, the increased open-field movement in the CNO group decreased back to baseline by day 11 (*Figure 3—figure supplement 1B*), just as calcium levels began to climb markedly (*Figure 3F*) and in parallel to axonal degeneration (*Figure 2A*, *Figure 2—figure supplement 1B*).

## Identifying activation-associated transcriptomic changes with region specificity using spatial transcriptomics

To gain additional insight into the mechanisms of degeneration in this model, we used spatial genomics. We performed Visium spatial transcriptomics on midbrain and striatal sections from DAT^IRIScre^ mice that were bilaterally injected with AAV-DIO-hM3Dq-mCherry at 3 months and then received either CNO (GqCNO) or vehicle (GqVeh) drinking water for 1 week prior to harvest. Additional non-injected control mice also received CNO (CNO alone) (*Figure 4A*). Mice were harvested following 1 week of CNO treatment in order to quantify transcriptomic changes at a time point when some axons still remain and prior to somatic degeneration (*Figure 2A*, *Figure 1—figure supplement 2A*, *Figure 4—figure supplement 1A*), thereby focusing on early gene expression changes prior to neuronal death.

We next identified 55 µm diameter barcoded discs (Visium Spatial Gene Expression) overlaying regions enriched for DA neurons within the SN and VTA in the midbrain, and the caudate putamen (CP) (*Figure 4B*). In the midbrain, selected discs contained at least one entire TH-positive (and mCherry-positive for injected animals) soma and also expressed at least two of three characteristic DA genes (*Slc6a3* which encodes the DA transporter DAT, *Slc18a2* which encodes the vesicular mono-amine transporter 2 [VMAT2], or *Th*) above a preset threshold level (*Figure 4C*). Interestingly, *Slc6a3* and *Slc18a2* were significantly decreased in the SN and VTA of GqCNO mice, perhaps indicating a loss of a dopaminergic phenotype. In agreement with this, we found that striatal DA levels were markedly decreased by HPLC (*Figure 4—figure supplement 2*). Principal components analyses of the SN and VTA in the midbrain demonstrate that GqCNO midbrain dopaminergic regions are transcriptionally similar to one another and distinct from control groups (*Figure 4D*). This suggests that chronic DREADD activation induces a robust transcriptomic response in DA neurons that is independent of viral transduction or CNO administration. Consistent with this, we employed the NEUROeSTIMator method (*Bahl et al., 2024*) to predict neural activity scores based on the transcriptomic assessment of the SN and VTA, finding that GqCNO samples had higher predicted activity scores than controls (GqVEH and CNO alone) (*Figure 4E*).

We hypothesized that chronic activation of DA neurons likely also leads to specific gene expression changes in striatal target neurons. In the striatum, discs were selected based on the expression of *Ppp1r1b* (which encodes DARPP32) (*Figure 4C*). Notably, principal components analysis between GqCNO and control groups revealed less separation in the striatum than the midbrain (*Figure 4D*), perhaps because of the multiple inputs on striatal neurons in addition to dopaminergic axons, although future investigation is required.

We next compared the transcriptomic profiles of discs within each brain region between GqCNO and GqVeh mice, as well as in GqCNO vs CNO alone mice. Only genes that had significant differential expression between GqCNO and both control groups were assessed for pathway changes (*Figure 4F*). Consistent with chronic DREADD activation, pathway analyses of GqCNO vs control groups in the SN and VTA showed significant enrichment of GO Biological processes, including 'chemical synaptic transmission', 'synaptic vesicle exocytosis', 'positive regulation of transporter activity', and 'vesicle transport along microtubule' (*Table 1*). Also consistent with DREADD activation, GO

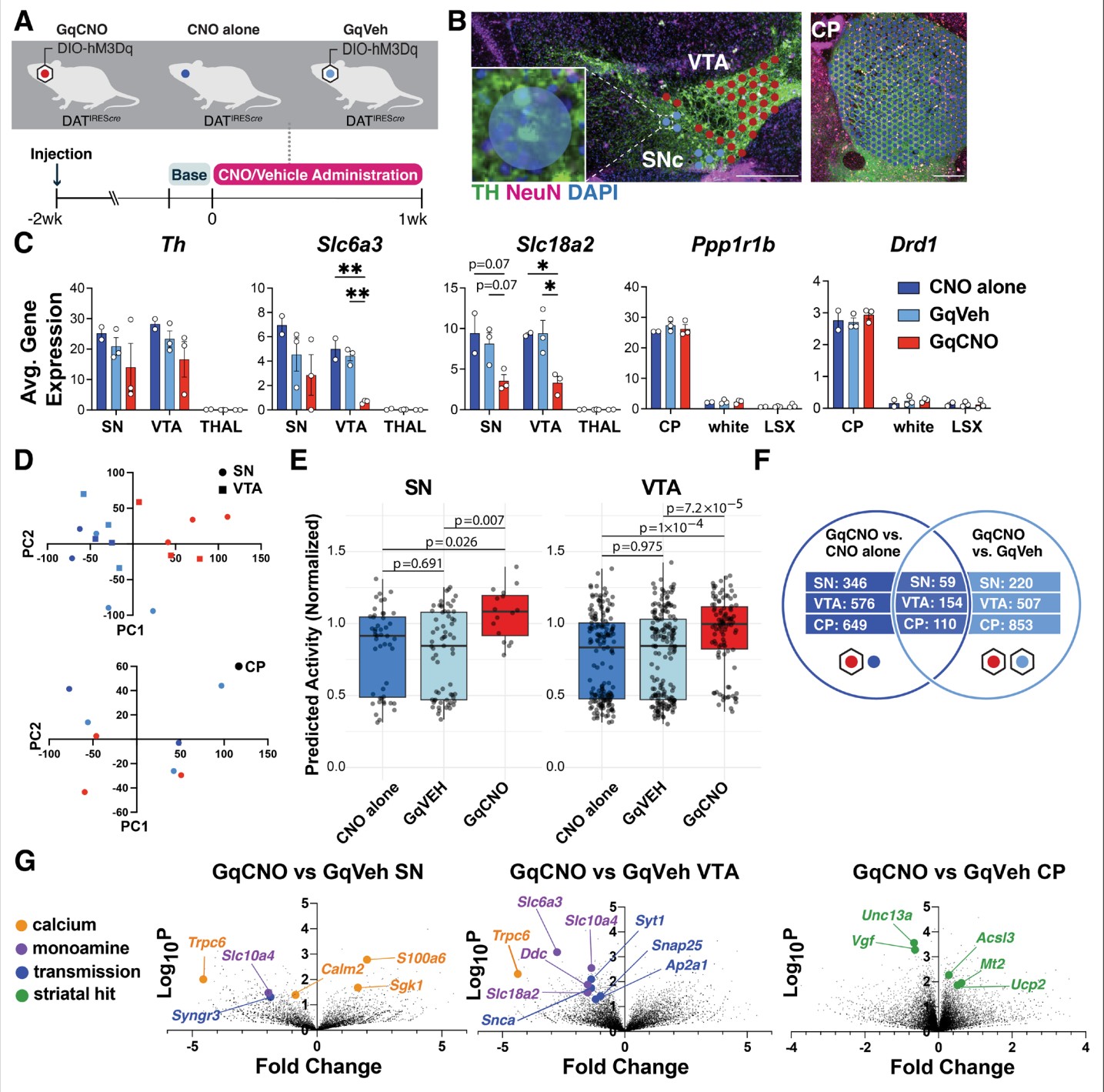

**Figure 4.** Spatial transcriptomics reveals midbrain dopamine (DA) and striatal target differentially expressed genes (DEGs) altered by chronic DA neuron hyperactivity. (**A**) DAT^IREScre^ animals that received clozapine-*N*-oxide (CNO) (CNO alone, n=2 mice) or were injected with AAV-hM3Dq-mCherry and received vehicle (GqVeh, n=3 mice) or CNO (GqCNO, n=3 mice) were treated for 1 week before brains were flash-frozen for spatial transcriptomic analysis. (**B**) Image of midbrain and striatal sections stained with TH (green), NeuN (purple), and DAPI (blue) shows discs assigned to regions of interest. Inset shows a disc containing two TH+ cell bodies. (**C**) Expression of dopaminergic and striatal genes is confined to expected spatial regions. Expression of genes involved in DA metabolism decreases with chronic CNO. 2–49 discs were compiled per ventral tegmental area (VTA), and 1–7 discs were compiled per SN. 357–560 capture areas were compiled per CP. The thalamus was selected as a midbrain control region, while white matter tracts (white) and the lateral septal complex (LSX) were used as striatal controls. (**D**) Principal components analysis of midbrain regions (top) and the caudate putamen (bottom) for GqCNO, GqVeh, and CNO alone groups. (**E**) The deep learning NEUROeSTIMator model was used to predict neural activity of GqCNO, GqVEH, and CNO alone within the SN and VTA from Visium spatial transcriptomics. Groups were compared using the Kolmogorov-Smirnov

*Figure 4 continued on next page*

*Figure 4 continued*

(KS) test (see Methods). (**F**) Hits were used for Enrichr pathway analysis if significant in both GqCNO vs CNO alone and GqCNO vs GqVeh comparisons. Gene rankings for hit analysis were established using fold change score (FCS) and signal-to-noise score (SNS). (**G**) Volcano plots comparing GqCNO vs GqVeh in the SN, VTA, and CP. Genes highlighted are also significantly altered when comparing GqCNO vs CNO alone. Scale bars indicate 500 µm. Data indicate mean ± SEM. *p<0.05, **p<0.01, ***p<0.001 by one-way ANOVA followed by Holm-Sidak post hoc test.

The online version of this article includes the following figure supplement(s) for figure 4:

**Figure supplement 1.** Additional spatial transcriptomic data following chronic hyperactivation.

**Figure supplement 2.** Striatal dopamine (DA) is decreased with chronic activation.

**Figure supplement 3.** Top differentially expressed genes in the mouse substantia nigra (SN) and ventral tegmental area (VTA).

**Figure supplement 4.** Top differentially expressed genes in the mouse caudate putamen (CP).

**Figure supplement 5.** Chronic isradipine treatment does not rescue axonal degeneration.

**Figure supplement 6.** Assessment of mouse differentially expressed genes (DEGs) in human Parkinson's disease (PD) and control substantia nigra pars compacta (SNc) samples.

Molecular Functions in the VTA included 'GTP binding' and 'syntaxin binding' (*Table 1*). Interestingly, both CP and SN GO Molecular Functions included 'calcium channel regulator activity', suggesting a larger circuit-level change in calcium channel regulation as a result of chronic DREADD activation (*Table 1*).

Consistent with this, there were a number of differentially expressed genes (DEGs) in the SN and/or VTA that regulate neural transmission, calcium, and activity (*Figure 4G*, *Figure 4—figure supplement 1B and C*, *Figure 4—figure supplement 3*, *Figure 4—figure supplement 4*). These included increased expression of *S100a6*, a calcium sensor and modulator (*Donato, 2003*) and *Sgk1*, a serine/threonine kinase regulated by intracellular calcium involved in the regulation of ion channels, membrane transporters, cellular enzymes, and transcription factors (*Lang et al., 2010*). Interestingly, *Sgk1* expression trended up in the CP, supporting circuit-level changes in calcium signaling. Increased *S100a6* expression has been observed in AD and ALS (*Filipek and Leśniak, 2020*), while *Sgk1* upregulation has been associated with DA neuron death in rodent toxin models of PD (*Iwata et al., 2004*; *Stichel et al., 2005*). Conversely, the expression of *calmodulin-2* (*Calm2*), which mediates control over a large number of enzymes and ion channels as a result of calcium binding (*Means et al., 1991*), and *Trpc6*, which is thought to form a nonselective calcium-permeant ion channel (*Hsu et al., 2007*), were both decreased. Decreased *Trpc6* may be a homeostatic response to chronically high intracellular calcium. These changes in calcium-regulating genes are consistent with the changes in baseline intracellular calcium observed in DA neurons by GCaMP photometry (*Figure 3*) and may reflect early processes that either cause or contribute to the onset of degeneration, or alternatively compensate to protect against degeneration.

Pathway-level changes in transmission were driven by decreases in several synaptic-vesicle genes, including *synaptotagmin-1*, a calcium sensor that triggers transmitter release including striatal DA release (*Xu et al., 2009*); *Snap25*, a t-SNARE that regulates transmitter release (*Kádková et al., 2019*); *Ap2a1*, a component of the adapter protein complex AP-2 (*Jackson et al., 2010*); and *Syngr3*, a synaptic vesicle-associated protein that may regulate DAT function (*Ho et al., 2023*). Interestingly, *Snca* was also significantly downregulated in the VTA. α-Synuclein is enriched in presynaptic terminals and has been shown to modulate DA release (*Bendor et al., 2013*). Together, these results suggest that chronic DREADD activation of DA neurons results in significant changes in synaptic transmission which may result in altered calcium regulation and DA release.

To begin to interrogate the role of calcium in hyperactivity-induced degeneration, we performed a pilot experiment in which mice injected with saline in the left midbrain and AAV-hM3Dq(DREADD)-mCherry in the right midbrain were then implanted subcutaneously with slow-releasing placebo or isradipine pellets (*Figure 4—figure supplement 5A*). Mice were not acutely tested for responsiveness to CNO, as wheel running data is less clear with unilateral manipulation. Instead, DREADD expression was validated post hoc by immunofluorescence, and animals with little or no DREADD expression were excluded from further analysis. Isradipine blocks $Ca_v1.2$ and $Ca_v1.3$ channels that are highly expressed in SNc neurons (*Chan et al., 2007*; *Dragicevic et al., 2014*). Isradipine treatment has been shown to return SNc DA neurons to a juvenile form of pacemaking, and chronic treatment has been shown to reduce calcium-dependent mitochondrial oxidant stress (*Chan et al., 2007*; *Guzman et al., 2018*). To

confirm the presence of systemic isradipine, we measured isradipine levels in blood plasma via HPLC (*Figure 4—figure supplement 5B*). Following 2 weeks of vehicle or CNO administration with placebo or isradipine release, we assessed striatal TH immunoreactivity and mCherry immunofluorescence. Consistent with previous results, we observed decreases in TH and mCherry in CNO-treated AAV-Gq hemispheres, but chronic isradipine administration did not rescue this loss (*Figure 4—figure supplement 5C*). However, the absence of rescue may result from a lack of statistical power due to small sample size or due to the presence of nonresponders. Further investigation is required to fully test the contribution of calcium signaling in hyperactivity-induced degeneration.

We also identified DEGs in the striatum. Upregulated genes included: *Mt2*, a metallothionein protein that acts as an antioxidant to protect against free radicals (*Carpenè et al., 2007*); *Acsl3*, an acyl-CoA synthetase that stimulates fatty acid synthesis (*Fernandez and Ellis, 2020*) and could supply energy substrates; and the mitochondrial uncoupling protein *Ucp2*. Downregulated genes included: *Unc13a*, which is involved in neurotransmitter release via vesicle maturation and has been implicated in ALS (*Dittman, 2019*; *Diekstra et al., 2014*), and *Vgf*, a secreted growth factor whose expression is also decreased in PD (*Quinn et al., 2021*).

We also evaluated the SN DEGs (n=59) and additional genes of interest identified within our DREADD mouse model with gene expression changes within human control and early PD subjects (*Figure 4—figure supplement 6A and B*). We employed the Nanostring GeoMx assay to assess whole transcriptome changes of DA neurons (TH+ masked, see Methods) within the SNc ventral tier of controls and early PD subjects. Of the 59 DEGs, 51 were found within the human dataset, and these genes exhibited broadly congruent directional changes between mouse and human (chi-square p-value=0.05). This suggests that similar mechanisms may be at play in the PD midbrain and the DREADD mouse model. To investigate this further, we considered genes that emerged from the mouse dataset involved in calcium, transmission, and DA and evaluated them in the early PD samples (*Figure 4—figure supplement 6B and C*). We found that the expression of *Syt1*, *Syngr3*, and *Calm2* was all decreased in both mouse and human datasets. Similarly, we found that expression of *Slc18a2* (VMAT2) and *Slc10a4* were also decreased in the mouse and human data. These findings may suggest common mechanisms of calcium dysregulation and altered DA metabolism in an early stage of degeneration in our mouse model and in human PD. Additionally, differential gene expression analysis revealed that *Hspa4* and *Eif3g* were downregulated (FDR<0.02) within the SNc ventral tier DA neurons of early PD subjects compared to controls and in the mouse model (*Figure 4—figure supplement 6A*), highlighting other potential common pathogenic mechanisms. Additional investigation is required to determine the functional impact of these genetic changes on DA neuron and DA neuron subtype function and vulnerability.

## Discussion

Here, we establish a new model system to chronically activate midbrain DA neurons and show that this prolonged activation leads to the preferential and stepwise degeneration of SNc DA neurons, starting with loss of terminals and progressing to neuronal death.

### New model to chronically activate DA neurons

Prior studies have activated midbrain DA neurons by both optogenetics and DREADDs to study the impact of DA neuron activity on sleep (*Eban-Rothschild et al., 2016*) and the spread of pathological α-synuclein (*Helwig et al., 2022*). However, in order to accurately model the impact of neural activity on neurodegeneration, we hypothesized that activity must be chronically increased. Unlike other studies which have used once or twice daily injections of CNO, we delivered CNO in the drinking water to ensure a more continuous, sustained impact on neural activity. We confirmed that chronic 1-, 2-, or 4-week hM3Dq activation results in sustained increases in intracellular calcium levels and functional behavioral readouts, suggesting that our model induces long-lasting changes in DA neuron activity. Moreover, ex vivo electrophysiology demonstrated that the basal level of DA neuron activity does indeed increase with CNO treatment. An important question, however, is whether our model accurately recapitulates how neural activity increases in physiological or pathological situations.

Compensatory increased or decreased rates of DA neuron pacemaking may help to maintain striatal DA as it is depleted in PD or to protect against energy failure by decreasing energy consumption.

Meanwhile, changes in bursting may reflect circuit-level adaptations to PD in response to ongoing degeneration. Irregular or silenced pacemaking is commonly observed in toxin-based models of PD (*Harden and Grace, 1995*; *Liss et al., 1999*; *Bilbao et al., 2006*; *Liss et al., 2005*; *Yee et al., 2014*), while increased induced bursting is observed after massive loss of DA neurons or complete loss of complex I (*Hollerman and Grace, 1990*; *Chen et al., 2009*; *Stachowiak et al., 1987*; *González-Rodríguez et al., 2021*). These changes could further promote the degeneration of remaining DA neurons. As such, in future studies, it will be important to understand how changes in the pattern of activity (i.e. bursting vs pacing) impact degeneration. In other PD models, DA neuron activity is altered prior to degeneration (*Bishop et al., 2010*; *Good et al., 2011*; *Lin et al., 2021*). The energetic demands of neural activity position altered activity as a possible contributor to the degenerative process in energetically vulnerable DA neurons, either as an initiating force or as a driver of disease progression.

One possible outcome of chronic hyperactivation is depolarization block (DB), a process that occurs even in healthy neurons wherein prolonged excitation induces persistent depolarization and blockade of spike generation (*Curtis et al., 1960*). DB occurs in midbrain DA neurons following chronic treatment with antipsychotic drugs (*Grace and Bunney, 1986*), and DB in SNc neurons is specifically associated with extrapyramidal symptoms (*Grace et al., 1997*; *White and Wang, 1983*). Lesions of DA neurons with 6-OHDA can also increase susceptibility to DB (*Hollerman and Grace, 1989*). In our chronic hyperactivation model, there is a strong decrease in dark cycle wheel running behavior that may be consistent with DB (*Figure 1C*, *Figure 1—figure supplement 1B*). Prolonged excitation along with progressive degeneration in our model may put DA neurons at greater risk of entering DB. However, CNO-treated animals have increased rather than decreased open-field movement (*Figure 3—figure supplement 1B*). Our NEUROeSTIMator data also predicts higher activity scores in GqCNO samples compared to GqVEH and CNO alone controls (*Figure 4E*). Moreover, while DB can't be assessed in slices due to its reliance on afferent input, we observed increased pacemaking in CNO-treated neurons, further arguing against prominent induction of DB. Given that VTA and SNc neurons can have differential susceptibilities to DB in response to antipsychotics, it is possible that a subpopulation of DA neurons in our model periodically enter DB without gross effects on locomotion. Additional data such as in vivo electrophysiology will be necessary to fully investigate the possibility of DB.

As expected, chronically increasing neural activity increased the movement of mice during the light cycle. However, it was notable that movement as measured by wheel use in the home cage did not increase as robustly as did movement in the open-field chamber (*Figure 3—figure supplement 1B*), perhaps indicating that the impact on movement is greater in novel environments. This may reflect the effects of midbrain DA neurons on motivation and exploration (*Wang et al., 2013*; *Jing et al., 2019*). The later return to control levels of locomotion midway through CNO treatment may indicate the onset of synaptic dysfunction or degeneration. In addition, our electrophysiologic analyses focused on cell body function, but the loss of terminals starting by ≈1 week (*Figure 2—figure supplement 1B*) raises the possibility that synaptic function might be disrupted even earlier, in parallel to behavioral changes. Meanwhile, chronically increasing activity unexpectedly and robustly decreased movement during the dark cycle (when mice should be most active) within 3–4 days of starting CNO (*Figure 1C*, *Figure 1—figure supplement 1B*). One possibility is decreased dark cycle movement results from a disruption in sleep, which might occur given the mild increased use of running wheels during the light cycle. Indeed, prior studies show that acute inhibition or stimulation of VTA DA neurons by optogenetics or DREADDs suppresses or promotes wakefulness, respectively (*Eban-Rothschild et al., 2016*; *Taylor et al., 2016*). These data support a critical role for VTA DA neurons in maintaining wakefulness and suggest that increased activity might promote wakefulness in our activity model. Future investigations assessing the impact of chronic SNc vs VTA DA neuron activation on circadian rhythms and sleep are needed to determine if these DA neurons similarly influence sleep in the setting of prolonged stimulation, and how potential sleep changes might contribute to the effects of chronically increased neural activity on motor function.

## Chronic chemogenetic activation drives degeneration

How does increasing the activity of SNc and VTA DA neurons impact these neurons, as well as their striatal targets? Analysis by spatial transcriptomics near the onset of axonal degeneration reveals a

clear decrease in the expression of genes involved in DA synthesis (*Th*), uptake (*Slc6a3*), and storage (*Slc18a2*) in the SNc and VTA, perhaps reflecting neuronal stress that precedes degeneration or an attempt to decrease DA release. Further investigation will be needed to determine whether these gene expression changes are associated with functional deficits. Decreases in presynaptic transmission-related genes (*Syt1*, *Snap25*, *Snca*, *Ap2a1*) in midbrain DA neurons are consistent with axonal dysfunction and loss that are observed later in CNO treatment. Moreover, consistent with the role of hM3Dq DREADDs in increasing cytosolic calcium (*Roth, 2016*) on a systems level, we observed a significant change in expression of genes involved in the regulation of calcium channels (*Table 1*), as well as serum and glucocorticoid-inducible kinase 1 (*Sgk1*), a protein-serine/threonine kinase activated by calcium (*Imai et al., 2003*), in both the midbrain and striatum. Our photometry analysis also raises the possibility of a central role for calcium in activity-induced degeneration, as calcium levels in midbrain DA levels rose markedly following 7 days of CNO (*Figure 3F*), as axonal degeneration progresses (*Figure 2—figure supplement 1B*).

The mechanism of preferential degeneration remains to be determined but is likely triggered by increased cytosolic calcium levels associated with hM3Dq activation (*Roth, 2016*). SNc DA neurons are believed to be intrinsically vulnerable to increased calcium due to their pacemaking activity driven by Ca$_v$1.3 channels (*Chan et al., 2007*), combined with the high energetic requirements of removing calcium from neurons and lack of calbindin expression (*Surmeier and Schumacker, 2013*). Further investigation is required to determine if DREADDs differentially increase calcium levels in SNc vs VTA DA neurons, and/or if SNc DA neurons are more vulnerable to the same level of elevation. Further, it remains unknown whether increased calcium leads directly to degeneration, for instance by triggering energy failure or increasing ROS, or if it indirectly causes neurodegeneration by triggering potentially toxic downstream processes such as increased cytosolic or extracellular DA release.

Excessive glutamate leads to large influxes of calcium that can trigger excitotoxicity leading to neurodegeneration and cell death (*Szydlowska and Tymianski, 2010*). Chronic, mild calcium elevation can also dysregulate mitochondrial calcium buffering and increase sensitivity to cytosolic calcium toxicity (*Verma et al., 2022*), and the route of calcium entry can also impact toxicity (*Sattler et al., 1998*). In a pilot study, isradipine, a blocker of Ca$_v$1.2 and Ca$_v$1.3 channels, did not confer protection against hyperactivity-induced degeneration in our model (*Figure 4—figure supplement 5*). However, it is possible that other interventions against calcium-induced toxicity, such as inhibition of NMDA receptors or the mitochondrial calcium uniporter, may be protective. Alternatively, degeneration could be driven by the toxicity of excessive extracellular DA as a consequence of elevated activity. Future studies will explore these potential mechanisms.

If increased DA neuron activity contributes to degeneration in PD, then one might also expect gene expression changes in our model to mirror those seen in PD. Indeed, we found that mouse DEGs exhibited broadly congruent directional changes in the human dataset (*Figure 4—figure supplement 6*). This raises the possibility that similar mechanisms of degeneration and adaptation may be at play in both our mouse model and in PD. Interestingly, *Hspa4* was significantly downregulated in the mouse and human datasets. Heat shock proteins broadly function to remove misfolded proteins, but the function of *Hspa4* in neurons is not well described. However, in rat neural stem cells, *Hspa4* is upregulated by selegiline, a type B monoamine oxidase inhibitor used to treat PD, and *Hspa4* upregulation reduces ROS levels and mitochondrial DNA damage following hydrogen peroxide exposure (*Abdanipour et al., 2018*; *Elkenani et al., 2022*). In MPTP-treated mice, deacetylation of *Hspa4* is linked to decreased microglial activation and neuroinflammation (*Yang et al., 2022*). In our activity model, decreased *Hspa4* expression might indicate a failure to respond to inflammation, promoting neurodegeneration. Further studies will be required to investigate the role of *Hspa4* and inflammation in chronic hyperactivation-induced toxicity.

Notably, we compared our mouse transcriptomic data at a stage when there is mild terminal loss and no somatic degeneration with early human PD patient tissues where there is limited DA neuron loss, raising confidence in the relevance of the comparison. However, it is important to consider that our Visium transcriptomic analysis has limited cell-type resolution. Therefore, important areas of future investigation include higher spatial resolution in the analysis of mouse tissue, and assessment of additional disease stages, brain areas, and cell types in human PD patient tissues.

In some cases, disease proteins or circuit changes may decrease neural activity. As such, it will be important to understand in which contexts activity increases or decreases, and also how decreasing

DA activity influences degeneration. Indeed, if activity does promote degeneration, it will also be important to understand if decreasing DA neuron activity is protective, and if activity can be decreased or modulated without compromising motor function. Interestingly, chronic nicotine, which may mediate the protective association of smoking in PD (*Quik, 2004*; *Nourse et al., 2021*), inhibits SNc DA neurons through agonism of nicotinic acetylcholine receptors expressed on presynaptic GABAergic terminals (*Xiao et al., 2009*). In addition, a proposed mechanism of action for the beneficial effects of deep brain stimulation (DBS) in PD is the inhibition of the subthalamic nucleus. As such, it will also be important to know how DBS influences DA neuron activity and if it has the potential to influence neurodegeneration. Interestingly, recent clinical trial data raise the possibility that DBS might slow disease progression when administered to early-stage patients (*Hacker et al., 2020*).

In summary, our data show that chronically increasing the activity of DA neurons can produce toxicity, and that SNc neurons may be more susceptible to this effect than VTA neurons. Considering that DA neuron activity may increase to compensate for other dying DA neurons, or in response to certain disease proteins, our data support the hypothesis that increased neural activity contributes to the pathophysiology of at least a subset of PD. Proving this, and determining how this evolves with disease stage, will be important goals for future research.

## Methods

### Key resources table

| Reagent type (species) or resource | Designation | Source or reference | Identifiers | Additional information |
|---|---|---|---|---|
| Strain, strain background (*Mus musculus, male*) | DAT*IREScre* | The Jackson Laboratory | RRID:IMSR_JAX:006660 | |
| Strain, strain background (*Mus musculus, male*) | Ai148D | The Jackson Laboratory | RRID:IMSR_JAX:030328 | |
| Antibody | Anti-tyrosine hydroxylase (rabbit polyclonal) | Millipore Sigma | RRID:AB_390204 | 1:1000 (IF, DAB), 1:100 (Visium) |
| Antibody | Anti-DsRed (rabbit monoclonal) | Takara | RRID:AB_3083500 | 1:1000 (IF, DAB) |
| Antibody | Anti-NeuN (mouse monoclonal) | Millipore Sigma | RRID:AB_2298772 | 1:200 (Visium) |
| Antibody | Anti-mCherry (chicken polyclonal) | Abcam | RRID:AB_2722769 | 1:200 (Visium) |
| Antibody | Anti-mouse Alexa Fluor 647 (goat polyclonal) | Thermo Fisher | RRID:AB_2535804 | 1:200 (Visium) |
| Antibody | Anti-rabbit Alexa Fluor 488 (goat polyclonal) | Thermo Fisher | RRID:AB_2576217 | 1:500 (IF), 1:200 (Visium) |
| Antibody | Anti-rabbit Alexa Fluor 594 (goat polyclonal) | Thermo Fisher | RRID:AB_2534095 | 1:500 (IF) |
| Antibody | Anti-chicken Alexa Fluor 594 (goat polyclonal) | Thermo Fisher | RRID:AB_2534099 | 1:200 (Visium) |
| Antibody | Anti-rabbit Biotinylated (goat polyclonal) | Vector Laboratories | RRID:AB_2313606 | 1:300 (DAB) |
| Recombinant DNA reagent | rAAV8-hSyn-DIO-hM3Dq-mCherry | UNC Vector Core, AddGene | RRID:Addgene_44361 | Diluted to 4.6e$^{12}$ mol/mL |
| Commercial assay or kit | Vectastain ABC-HRP Kit, Peroxidase (Rabbit IgG) | Vector Laboratories | PK-4001 | |
| Commercial assay or kit | Visium Spatial Gene Expression (v1) | 10x Genomics | 1000215, 1000187, 1000193, 1000200 | |
| Commercial assay or kit | GeoMx Digital Spatial Profiler | Nanostring | RRID:SCR_021660 | |
| Chemical compound, drug | Clozapine-*N*-oxide | Tocris Bioscience | 4936 | 300 mg/L via drinking water |

*Continued on next page*

*Continued*

| Reagent type (species) or resource | Designation | Source or reference | Identifiers | Additional information |
|---|---|---|---|---|
| Chemical compound, drug | Isradipine slow-release pellets | Innovative Research of America | Custom pellets; isradipine supplied by Thermo Fisher (J63920.MF) | 3 µg/g/day release |
| Software, algorithm | EthoVision XT | Noldus | RRID:SCR_000441 | Version 10 |
| Software, algorithm | Wheel Analysis Software | Med Associates | SOF-861 | |
| Software, algorithm | Enrichr | *Sei et al., 2023*; *Chen et al., 2013*; *Kuleshov et al., 2016* | RRID:SCR_001575 | |
| Software, algorithm | Space Ranger | 10x Genomics | RRID:SCR_023571 | |
| Software, algorithm | Loupe Browser | 10x Genomics | RRID:SCR_018555 | Version 5 |
| Software, algorithm | Stereo Investigator | MBF Bioscience | RRID:SCR_024705 | |
| Software, algorithm | Synapse | TDT | RRID:SCR_006307 | |
| Software, algorithm | GeoMx NGS analysis pipeline | Illumina | RRID:SCR_011881 | Version 2.0.21 |

## Experimental model and subject details

### Mice

Mice were group-housed in a colony maintained with a standard 12 hr light/dark cycle and given food and water ad libitum. All mice received food on the cage floor. All animal experimental procedures were conducted in accordance with the Guide for the Care and Use of Laboratory Animals, as adopted by the National Institutes of Health, and with approval from the University of California, San Francisco Institutional Animal Care and Use Committee. All mice were housed in a state-of-the-art barrier facility managed by the UCSF Laboratory Animal Resource Center (LARC). Animal care and use in this research are covered under the UCSF 'Assurance of Compliance with PHS Policy on Humane Care and Use of Laboratory Animals by Awardee Institutions' number A3400-01. Experiments were performed on age-matched mice. All mouse lines were maintained on a C57Bl/6 background (The Jackson Laboratory; RRID:IMSR_JAX:000664). DAT$^{IREScre}$ mice (*Bäckman et al., 2006*) and Ai148D mice (*Daigle et al., 2018*) were obtained from The Jackson Laboratory.

## Methods

### Chemogenetics

A detailed protocol for mouse stereotaxic surgery can be found at https://dx.doi.org/10.17504/protocols.io.b9kxr4xn. DAT$^{IREScre}$ homozygous mice (RRID:IMSR_JAX:006660) were injected bilaterally with 1 µL of rAAV8-hSyn-DIO-hM3Dq-mCherry (UNC Vector Core, AddGene, RRID:Addgene_44361) into the midbrain (–3.0 mm anterior-posterior, ±1.2 mm medial-lateral, –4.3 mm dorsal-ventral) using a stereotaxic frame (Kopf) and a microliter syringe (Hamilton).

A detailed protocol for monitoring mouse activity can be found at https://dx.doi.org/10.17504/protocols.io.3byl4qqe8vo5/v1. Two weeks following surgery, animals were single-housed and habituated to wireless running wheels (MedAssociates ENV-047, RRID:SCR_024879) connected to a hub (MedAssociates DIG-807, RRID:SCR_024880) for locomotion recordings and water bottles (Amazon) for drinking. A detailed protocol for validating responsiveness of DREADD-injected DAT$^{IREScre}$ mice to CNO can be found at https://dx.doi.org/10.17504/protocols.io.6qpvr3yrpvmk/v1. Validation of responsiveness was done for all DREADD experiments, except for those unilaterally injected for chronic isradipine treatment (*Figure 4—figure supplement 5*). All mice had access to a running wheel during their treatment, except those for fiber photometry and chronic isradipine treatment. The mice used for photometry underwent 7 days of running wheel access approximately 3 weeks prior to the beginning of the experiment. The photometry headcaps sterically prevented mice from having access to a running wheel in their home cage. Mice used for chronic isradipine treatment were unilaterally injected with hM3Dq, which does not lend to clear interpretation of running wheel data.

**Table 1.** Pathway analyses for the ventral tegmental area (VTA), substantia nigra (SN), and caudate putamen (CP) following chronic hyperactivation.

Gene Ontology Molecular Function and Biological Process terms for differentially expressed genes in the VTA, SN, and CP generated with the Enrichr webtool.

**VTA**

| Index | GO Molecular Function 2023 | p-Value |
|---|---|---|
| 1 | GTP Binding (GO:0005525) | 0.00002637 |
| 2 | Syntaxin Binding (GO:0019905) | 0.00005402 |
| 3 | Guanyl Ribonucleotide Binding (GO:0032561) | 0.00006568 |
| 4 | Nuclear Receptor Coactivator Activity (GO:0030374) | 0.0005911 |
| 5 | Purine Ribonucleoside Triphosphate Binding (GO:0035639) | 0.001174 |

| Index | GO Biological Process 2023 | p-Value |
|---|---|---|
| 1 | Chemical Synaptic Transmission (GO:0007268) | 0.00000141 |
| 2 | Synaptic Vesicle Exocytosis (GO:0016079) | 0.0000152 |
| 3 | Anterograde Trans-Synaptic Signaling (GO:0098916) | 0.00002437 |
| 4 | Response To Cytokine (GO:0034097) | 0.00005223 |
| 5 | Response To Interferon-Beta (GO:0035456) | 0.00006911 |

**SN**

| Index | GO Molecular Function 2023 | p-Value |
|---|---|---|
| 1 | Phosphatase Activator Activity (GO:0019211) | 0.001131 |
| 2 | Tubulin Binding (GO:0015631) | 0.002579 |
| 3 | Calcium Channel Regulator Activity (GO:0005246) | 0.005332 |
| 4 | Microtubule Binding (GO:0008017) | 0.005412 |
| 5 | Alpha-Tubulin Binding (GO:0043014) | 0.005618 |

| Index | GO Biological Process 2023 | p-Value |
|---|---|---|
| 1 | Positive Regulation Of Transporter Activity (GO:0032411) | 8.673E-06 |
| 2 | Establishment Of Spindle Orientation (GO:0051294) | 0.0001492 |
| 3 | Neuron Projection Morphogenesis (GO:0048812) | 0.0009083 |
| 4 | Long-Term Memory (GO:0007616) | 0.001571 |
| 5 | Vesicle Transport Along Microtubule (GO:0047496) | 0.001903 |

**CP**

| Index | GO Molecular Function 2023 | p-Value |
|---|---|---|
| 1 | Protein Phosphatase 2A Binding (GO:0051721) | 0.007033 |
| 2 | Histone Deacetylase Binding (GO:0042826) | 0.01734 |
| 3 | Calcium Channel Regulator Activity (GO:0005246) | 0.01761 |
| 4 | Protein Phosphatase Binding (GO:0019903) | 0.02563 |
| 5 | Arachidonate-CoA Ligase Activity (GO:0047676) | 0.0272 |

| Index | GO Biological Process 2023 | p-Value |
|---|---|---|
| 1 | DNA Deamination (GO:0045006) | 0.001908 |
| 2 | Positive Regulation Of Mitotic Cell Cycle Phase Transition (GO:1901992) | 0.005059 |
| 3 | Positive Regulation Of G2/M Transition Of Mitotic Cell Cycle (GO:0010971) | 0.00588 |

*Table 1 continued on next page*

*Table 1 continued*

**VTA**

| | | |
|---|---|---|
| 4 | Positive Regulation Of Cell Cycle G2/M Phase Transition (GO:1902751) | 0.007645 |
| 5 | Secondary Alcohol Biosynthetic Process (GO:1902653) | 0.007645 |

A detailed protocol for preparing and administering CNO via drinking water can be found at dx.doi.org/10.17504/protocols.io.n2bvj33oblk5/v1. CNO (NIMH, Tocris 4936) was administered ad libitum in 2% sucrose water at 300 mg/L. CNO and vehicle waters were made fresh weekly and stored at 4°C protected from light.

### Ex vivo recording

A detailed protocol for ex vivo electrophysiology can be found at dx.doi.org/10.17504/protocols.io.261gedqn7v47/v1. DAT$^{IREScre}$ mice injected bilaterally with rAAV8-hSyn-DIO-hM3Dq-mCherry and chronically treated with vehicle or CNO water for 1 week were provided to the electrophysiologist blind to in vivo treatment. Mice were deeply anesthetized with isoflurane, decapitated with a rodent guillotine, and the brains were removed. Horizontal brain slices (150 µm thick) containing the SNc were prepared using a Vibratome (Campden Instruments, 7000smz-2). Slices were cut in ice-cold aCSF solution containing (in mM): 119 NaCl, 2.5 KCl, 1.3 $MgSO_4$, 1.0 $NaH_2PO_4$, 2.5 $CaCl_2$, 26.2 $NaHCO_3$, and 11 glucose saturated with 95% $O_2$–5% $CO_2$ and allowed to recover at 33°C in aCSF for at least 1 hr.

Slices were visualized under an *Axio Examiner A1* equipped with Dodt and IR optics, using a Zeiss Axiocam 506 mono and Neurolucida 2023 (MBF Biosciences) software. Neurons were identified as DREADD-expressing prior to patching with fluorescent imaging of mCherry expressed concurrently with DREADDs. Whole-cell patch-clamp recordings were performed at 33°C using 3–5 MΩ pipettes containing (in mM): 123 K-gluconate, 10 HEPES, 0.2 EGTA, 8 NaCl, 2 MgATP, and 0.3 $Na_3GTP$, pH 7.2, osmolarity adjusted to 275. Biocytin (0.1%) was also included in the internal solution to identify neurons after recordings where desired.

Recordings were made using Sutter IPA and SutterPatch v2.3.1 software (Sutter Instruments), filtered at 5 kHz and collected at 10 kHz. Liquid junction potentials were not corrected. $I_h$ was recorded by voltage clamping cells at –60 mV and stepping to –40,–50, −70,–80, −90,–100, −110, and –120 mV. Cells were recorded in current-clamp mode (*I*=0 pA) for measuring spontaneous firing rates and CNO responsivity. For CNO testing, spontaneous firing rate or resting membrane potential was monitored until a stable baseline was observed for at least 5 min. Then, the perfusion solution was switched to 1 µM CNO for 5 min.

When recordings were completed, slices were drop-fixed in 4% formaldehyde in PBS for at least 2 hr.

For quantifications, spontaneous firing rate was measured as the mean firing rate during the first 120 s of whole-cell recording. Once every 10 s a brief hyperpolarizing pulse was applied, and the input resistance of the neuron was quantified from this test, averaged across the measurements made during the first 2 min of recording. AP waveform measurements were made from averages across at least eight APs from this recording interval. $I_h$ magnitude was quantified as the difference between the initial steady-state response to the –120 mV step and the asymptote of the slow current sag.

**Table 2.** Demography of the human postmortem cohort assayed by GeoMx.

| Group | Gender (M/F) | Age at death (years) | Postmortem delay (hr) | DV200 |
|---|---|---|---|---|
| Aged healthy controls | 3/7 | 92.0 (5.75) | 25 (8.25) | 28.9 (16.4) |
| Early PD | 5/3 | 73.5 (12)[†] | 17.5 (13.5)* | 28.3 (9.10) |

Values are presented as median (IQR). The comparison of age at death, postmortem delay, and DV200 between groups was made using the Welch's two-sample t-test. The comparison of gender between groups was made using the chi-square test.
*p<0.05.
[†]p<0.001.

Statistical analyses were completed in R (4.2.3), first testing whether data met the criteria for parametric statistical evaluation. Those datasets that met criteria were compared by unpaired Student's t-test. Those that did not meet these criteria were compared by permutation test. Code for the electrophysiology analysis can be found at https://doi.org/10.5281/zenodo.10819038.

All salts and reagents were purchased from Sigma, except CNO (Tocris).

## Ex vivo data analysis

Results are presented as mean ± SEM or with kernel density estimations. All but one neuron recorded in current clamp was quiescent during CNO testing. This neuron both depolarized and increased its firing rate in response to CNO and was therefore included in the time course average figure. For all data, parametric assumptions were tested to choose between t-test (parametric) and permutation (nonparametric) analysis.

## Fiber photometry

A detailed protocol for collecting fiber photometry data can be found at dx.doi.org/10.17504/protocols.io.bp2l6xwxzlqe/v1. 3-month-old Ai148D mice (RRID:IMSR_JAX:030328) bred with DAT$^{IREScre}$ were injected with rAAV8-hSyn-DIO-hM3Dq-mCherry as described above. After allowing 3 weeks for expression of the hM3Dq construct, locomotion was tested using IP injection of CNO to determine robust hM3Dq expression. Mice that exhibited increased locomotion following CNO injection were then implanted with optical fibers (400 µM, 0.48 NA). Mice were administered vehicle or CNO water for 14 days.

In vivo calcium data was acquired using a custom-built photometry system. An RZ5P fiber photometry processor (TDT, RRID:SCR_024878) and Synapse software (TDT, RRID:SCR_006307) were used to control LED output and acquire the photometry signal. Using this system, two LEDs were used to control GCaMP and isosbestic excitation (470 nm and 405 nm, respectively, Thorlabs). LEDs were sinusoidally modulated at 211 Hz (470 nm) and 531 Hz (405 nm) and entered a four-port fluorescence mini cube (Doric Lenses). The combined output was coupled to a fiber-optic patch cord (400 µm, 0.48 NA, Thorlabs), which then mated to the fiber-optic cannula in the mouse brain. The emitted light was collected onto a visible femtowatt photoreceiver module (AC low, Newport) and sampled at 60 Hz. Photometry data was then extracted via proprietary TDT software using MATLAB (MathWorks, RRID:SCR_001622). Code for the fiber photometry analysis can be found at https://doi.org/10.5281/zenodo.10819072.

## Open-field behavior

A detailed protocol for open-field analysis can be found at dx.doi.org/10.17504/protocols.io.36wgq33rklk5/v1. Spontaneous locomotor activity was assessed while simultaneously recording fiber photometry data. Videos acquired during photometry sessions were analyzed using Ethovision software (Noldus, RRID:SCR_000441) to calculate total distance traveled and percent of time spent moving.

## Chronic isradipine treatment and plasma detection

A detailed protocol for implantation of slow-release pellets can be found at dx.doi.org/10.17504/protocols.io.n92ldr428g5b/v1. DAT$^{IREScre}$ mice were unilaterally injected in the right midbrain with AAV-hM3Dq as detailed above and in the left midbrain at the same coordinates with 1 µL of sterile saline at 200 nL per minute. Mice were allowed to heal for 2 weeks. Mice were then implanted with either placebo or isradipine slow-release pellets (Innovative Research of America). Pellets were designed to release 3 µg/g/day of isradipine. Briefly, mice were anesthetized with isoflurane, and a placebo or isradipine pellet was implanted subcutaneously, and the wound was closed with sutures. Administration of either vehicle or CNO water was initiated the same day as pellet implantation. Blood was drawn from anesthetized mice and incubated in tubes containing K3 EDTA solution on ice. Tubes were centrifuged at 4°C at 10,000×$g$ for 10 min. Collected supernatant was stored at –80°C until analyzed. Samples were sent to Charles River Laboratories for LC-MS/MS bioanalysis.

## Monoamine quantification

A detailed protocol for collecting striatal punches can be found at dx.doi.org/10.17504/protocols.io.dm6gp97xdvzp/v1. Striatal tissues flash-frozen for Visium Spatial Transcriptomics (described below)

were used for monoamine quantification. Briefly, flash-frozen brains were embedded in optimal cutting temperature (OCT) and mounted on a cryostat to collect dorsal striatal punches. Punches were stored at –80°C before shipment to the Vanderbilt Neurochemistry Core for catecholamine quantification by HPLC (*Berthet et al., 2014*).

## Immunohistochemistry

Animals were anesthetized with 2,2,2-tribromoethanol and perfused with PBS followed by 4% para-formaldehyde (PFA) in PBS. Intact brains were removed, post-fixed in 4% PFA overnight at 4°C, and cryoprotected in 30% sucrose. 40-μm-thick coronal sections were cut on a sliding microtome (Leica) and stored in cryoprotectant (30% ethylene glycol [Sigma], 30% glycerol [Fisher Scientific] in PBS).

A detailed protocol for immunofluorescence can be found at dx.doi.org/10.17504/protocols.io. kxygx38owg8j/v1. Brain sections were rinsed with PBS followed by 0.2% Triton X-100 in PBS. The sections were then transferred to a blocking solution containing 4% fetal bovine serum (JR Scientific) and 0.2% Triton X-100 in PBS for 1 hr. Following overnight incubation in primary antibody, sections were rinsed in 0.2% Triton X-100 in PBS and incubated for 2 hr with a suitable secondary antibody. Sections were rinsed again in 0.2% Triton X-100 before mounting and coverslipping with antifade mounting medium (Vector Laboratories H1400, H1500).

A detailed protocol for peroxidase staining can be found at dx.doi.org/10.17504/protocols.io. n92ldm127l5b/v1. Sections were quenched with 3% $H_2O_2$ and 10% methanol in PBS and blocked in 10% fetal bovine calf serum, 3% BSA, and 1% glycine in PBS with 0.5% Triton X-100. They were incubated with primary antibody followed by biotinylated secondary and subsequently streptavidin-conjugated horseradish peroxidase (HRP) (1:500; Vectastain ABC kit, Vector Laboratories). Immunostaining was visualized with hydrogen peroxide and 1 3,3′-diaminobenzidine (DAB, Sigma).

The following primary antibodies were used: rabbit anti-DsRed (1:1000; Takara, RRID:AB_3083500), rabbit anti-tyrosine hydroxylase (1:1000; AB152, Millipore, RRID:AB_390204), mouse anti-NeuN (Millipore MAB 377, RRID:AB_2298772), and chicken anti-mCherry (Abcam ab205402, RRID:AB_2722769). Secondary antibodies Alexa Fluor goat anti-mouse 647 (Thermo Fisher Scientific Cat# A-21235, RRID:AB_2535804), anti-rabbit 488 (Thermo Fisher Scientific Cat# A11034, RRID:AB_2576217), anti-rabbit 594 (Thermo Fisher Scientific Cat# A11037, RRID:AB_2534095), or anti-chicken 594 (Thermo Fisher Scientific Cat# A-11042, RRID:AB_2534099) IgG were used (1:500). A biotinylated goat anti-rabbit IgG (1:300, Vector Laboratories, BA-1000, RRID:AB_2313606) was used for peroxidase staining.

Stained samples were visualized using an automated fluorescence microscope (Keyence BZ7000) and a laser-scanning confocal microscope (Zeiss LSM880).

## Stereology

Total numbers of TH-positive and mCherry-positive neurons were quantified by an experimenter blinded to groups. Region selection of SN and VTA was done under ×5 magnification following the definition by *Fu et al., 2012*. Imaging was done under ×63 magnification by a Zeiss Imager microscope (Carl Zeiss Axio Imager A1) equipped with an XYZ computer-controlled motorized stage and an EOS rebel T5i Digital Camera (Canon), and counting was done using MBF Bioscience's Stereo Investigator (RRID:SCR_024705). Counting frame size was 60×60 μm$^2$, and systematic random sampling grid was 130×130 μm$^2$, with a section interval of 6.

## Optical density analysis

A detailed protocol for optical density analysis can be found at dx.doi.org/10.17504/protocols.io. 81wgbxo2nlpk/v1. Images of DAB-stained striatal sections were taken with an automated light microscope (Keyence) by a blinded experimenter. ImageJ (RRID:SCR_002285) was calibrated for optical density and subsequently used to draw regions of interest (ROIs) around striatal areas and to measure mean optical density. The Allen Brain Atlas was used as a reference brain atlas (Allen Mouse Brain Atlas, mouse.brain-map.org and atlas.brain-map.org). Optical density values were background-subtracted using an adjacent brain region with low TH expression levels, the piriform cortex.

## Spatial genomics

### Mouse Visium spatial gene expression

Spatial transcriptomics were acquired with Visium spatial gene expression kits (10x Genomics). Sample preparation, sample imaging, and library generation were completed in accordance with 10x Spatial Gene Expression protocols and as previously published (*Sei et al., 2023*). Briefly, fresh brain tissue was flash-frozen in an isopentane bath cooled to –80°C using dry ice. The brain tissue was then embedded in OCT compound (Tissue-Tek 62550-12). A cryostat was used to obtain a 10-μm-thin section from the midbrain that was then mounted onto a 10× spatial gene expression slide. Sections were stained with TH, NeuN, and DsRed to visualize mCherry (striatal sections only) and Hoechst 33342 before imaging on a Leica Aperio Versa slide scanner. The cDNA libraries were generated at the Gladstone Genomics Core. Libraries were sequenced at the UCSF Center for Advanced Technology on an Illumina NovaSeq 600 on an SP flow cell. Alignment of the sequencing data with spatial data from the Visium slide was completed with the 10x Space Ranger software (10x Genomics, RRID:SCR_023571). The 10x Loupe Browser software (RRID:SCR_018555) was then used to identify six anatomical ROIs: (1) SN, (2) VTA, (3) thalamus in the midbrain, and (4) CP, (5) LSX, and (6) white matter tracts in the striatum. RNA capture areas corresponding to each anatomical region were selected for analysis based on their spatial proximity to the anatomical regions and on the expression of known genetic markers. SN and VTA genetic markers included *Th*, *Slc6a3*, and *Slc18a2*; thalamus markers included *Prkcd*, *Ptpn3*, and *Synpo2*; CP was identified with *Ppp1r1b*; the LSX was identified with *Prkcd*; white matter was identified with *Mbp*. Demarcation of SN and VTA was done according to *Fu et al., 2012*. Capture areas that expressed high levels of astrocyte markers (*Gfap*, *Aldh1l1*) and microglia markers (*Aif1*, *P2ry12*) were excluded. SN and VTA capture areas also had to contain at least one complete DA neuron soma. Roughly 5–10 neuronal cell bodies fit into a single capture area for SN and VTA with 2–49 capture areas per VTA, 1–7 capture areas per SN, 0–39 capture areas per thalamus, 357–560 capture areas per CP, 6–93 capture areas per LSX, and 21–72 capture areas per white matter tracts. Gene expression levels were exported to GraphPad Prism and Microsoft Excel (Microsoft, RRID:SCR_016137). Gene rankings for hit analysis were established using the fold change score (FCS) and signal-to-noise score (SNS). Equations for these scores are given as:

$$SNS = \frac{\mu_{KO} - \mu_{WT}}{\sigma_{KO} + \sigma_{WT}} \times \left(-Log_{10}P\right)$$

$$FCS = Log_2 \left(\frac{\mu_{KO}}{\mu_{WT}}\right) \times \left(-Log_{10}P\right)$$

where μ is the average gene expression, σ is the standard deviation, and P is the p-value derived from a t-test. Genes with a p-value<0.05 were highlighted as DEGs of interest. The expression levels for these genes of interest were probed in SN, VTA, and the thalamus to identify a subset of genes with DA region-specific changes. Pathway analysis was done on all hits that appeared in both scoring metrics using the Enrichr webtool (RRID:SCR_001575) (*Chen et al., 2013*; *Kuleshov et al., 2016*; *Xie et al., 2021*). Code for Visium spatial transcriptomics analysis can be found at https://doi.org/10.5281/zenodo.10819193.

### Principal components analysis

The R function prcomp was used for principal component analysis, with the median normalized gene expression level of each gene as the input. Only genes that were expressed in all regions (for SN and VTA) and whose expression was >0 were included.

### Neuronal activity predictions

To quantify neuronal activity, we utilized NEUROeSTIMator (*Bahl et al., 2024*), a deep learning model that generates a singular activity score following whole transcriptome reconstruction of 22 neuronal activity markers. This score has been shown to correlate significantly with electrophysiological features of hyper-excitability. We applied the NEUROeSTIMator algorithm to the Visium count matrix (species = 'mmusculus'). Visium spatial transcriptomic capture areas within the SN and VTA were defined as described above, and neighboring spots were included using the RegionNeighbors (mode = 'all_inner_outer') function of semla R statistical program (*Larsson et al., 2023*). The predicted activity

scores were normalized by division of the mean activity scores of the total tissue section they were derived from. Group-wise normalized predicted activity scores were compared using the Kolmogorov-Smirnov test (KS test).

## Human spatial gene expression

Cohort materials: formalin-fixed paraffin-embedded (FFPE) midbrain sections were collected from individuals with pathologically verified early-stage Parkinson's disease (ePD) (n=10) and individuals without any neurological or neuropathological conditions (n=10) through the New South Wales (NSW) Brain Banks, as detailed in *Table 2*. The research protocol received ethical clearance from the University of Sydney Human Research Ethics Committee (approval number 2021/845). All PD subjects exhibited a positive response to levodopa and met the UK Brain Bank Clinical Criteria for PD diagnosis (*Gibbels, 1988*) without any other neurodegenerative disorders. Control subjects demonstrated no signs of Lewy body pathology, and ePD subjects had Braak stage IV Lewy body pathology, in accordance with the established criteria (*Braak et al., 2003*; *Halliday et al., 2006*).

Sequencing and data processing: FFPE sections were stained using TH antibody (BioLegend, cat# 818008, 1:50 RRID:AB_2801155) and processed using the Nanostring GeoMx Digital Spatial Profiler (RRID:SCR_021660), as per the manufacturer's instructions, to obtain TH+ masked transcripts. Sequencing libraries for the whole transcriptome were constructed using the Human Whole Transcriptome Atlas (GeoMx Hu WTA) following the manufacturer's instructions. Technical replicates were consolidated using the Linux 'cat' command, and alignment and feature counting were performed using the GeoMx NGS analysis pipeline (version 2.0.21) executed on the Illumina BaseSpace platform (Illumina, RRID:SCR_011881). Quality control was implemented using R statistical programming language with the cutoffs; minSegmentReads: 1000, percentTrimmed: 80%, percentStitched: 80%, percentAligned: 75%, percentSaturation: 50%, minNegativeCount: 1, maxNTCCount: 10,000, minNuclei: 20, minArea: 1000. The minimum gene detection rate across all samples was set at 1%. The minimum gene detection rate per sample was set to 1%. Recently, Van Hijfte and colleagues reviewed the recommended Nanostring GeoMx Q3 normalization technique (*van Hijfte et al., 2023*), observing that Q3 normalization failed to correct for large differences in the signal (gene expression) to noise (neg probes) observed between samples and recommended quantile normalization. We performed preliminary experiments of five normalization techniques: Q3 normalization, background normalization, quantile normalization, SCTransform, and normalizing for total library size. We observed that quantile normalization displayed the lowest absolute MA plot correlation and least significance following the KS test. Hence, quantile normalization was implemented, and batches (slides) were corrected using Harmony (*Korsunsky et al., 2019*). DEGs identified by spatial transcriptomic analysis of mouse model were evaluated between control and ePD TH+ masked ROIs from the SNc ventral tier. The congruence between directional changes of mouse and human was quantified using a chi-square test. To assess differential gene expression between control (CTR) and PD samples, we employed the 'limma voom' methodology, incorporating covariates diagnosis, age, postmortem delay, RNA integrity number equivalent (DV200), sex, DSP processing plate, and brain bank ID (*Law et al., 2014*). This analysis was conducted using the R statistical software environment with all scripts available at https://doi.org/10.5281/zenodo.10819064 and processed and raw datasets available at https://doi.org/10.5281/zenodo.10511809.

## Quantification and statistical analysis

All statistical analyses including the n, what n represents, description of error bars, statistical tests used, and level of significance are found in the figure legends or corresponding results. A minimum sample size of n=3 mice was used, except for spatial transcriptomics where capture area space was limited. Two-way repeated-measures ANOVA followed by Holm-Sidak multiple comparisons was used for comparing vehicle vs CNO groups. One-way ANOVA followed by Holm-Sidak multiple comparisons was used for comparing multiple groups in mouse spatial transcriptomic data. Three-way ANOVA followed by Holm-Sidak multiple comparisons was used for assessing chronic isradipine histology. t-Test or permutation (nonparametric) analysis was used for ex vivo electrophysiology data. Mouse DEGs were ranked according to the defined SNS and FCS metrics. Congruence between directional changes of mouse and human was quantified using a chi-square test. 'Limma voom' methodology was used to assess differential gene expression between control and PD samples, incorporating covariates

(see Human Spatial Gen Expression). All analyses were performed using GraphPad Prism version 9 (RRID:SCR_002798), Microsoft Excel, and R version 4.2.2 (RRID:SCR_000432).

## Acknowledgements

We acknowledge Haru Yamamoto and James Maas for assistance in preparing the manuscript, and Kathryn Claiborn for helping edit the manuscript and Erica Delin for administrative assistance. We thank Saheli Singh for providing assistance with mouse spatial transcriptomics analysis, Robert Edwards, Zayd Khaliq, Talia Lerner, and Christopher Ford from 'Team Edwards' for feedback on experiments, and Yutau Liu for technical assistance. This research was funded in whole or in part by Aligning Science Across Parkinson's (ASAP-020529, KN, GH, AN) through the Michael J Fox Foundation for Parkinson's Research (MJFF). For the purpose of open access, the author has applied a CC BY public copyright license to all Author Accepted Manuscripts arising from this submission. This work was also supported by NIH (RO1NS091902 and RF1AG064170, KN; R01DA030529, EBM; F31NS137765, KR), the State of California for medical research on alcohol and substance abuse through the State of California, the Joan and David Traitel Family Trust and Betty Brown's Family, a Burroughs-Wellcome Fund Award (KN), and the Gladstone Institutes. This work was also supported by the Hillblom Foundation (ZD, DH) and a Berkelhammer Award for Excellence in Neuroscience (ZD).

## Additional information

### Funding

| Funder | Grant reference number | Author |
|---|---|---|
| Aligning Science Across Parkinson's | ASAP-020529 | Glenda Halliday<br>Alexandra B Nelson<br>Ken Nakamura |
| National Institutes of Health | RO1NS091902 | Ken Nakamura |
| National Institutes of Health | RF1AG064170 | Ken Nakamura |
| National Institutes of Health | R01DA030529 | Elyssa Margolis |
| National Institutes of Health | F31NS137765 | Katerina Rademacher |
| California Department of Alcohol and Drug Programs | | Elyssa Margolis |
| Gladstone Institutes | | Ken Nakamura |
| Burroughs Wellcome Fund | | Ken Nakamura |
| Larry L. Hillblom Foundation | | Zak Doric<br>Dominik Haddad |

The funders had no role in study design, data collection and interpretation, or the decision to submit the work for publication.

### Author contributions

Katerina Rademacher, Data curation, Formal analysis, Supervision, Funding acquisition, Validation, Investigation, Visualization, Methodology, Writing – original draft, Writing – review and editing; Zak Doric, Data curation, Formal analysis, Supervision, Funding acquisition, Investigation, Visualization, Methodology, Writing – original draft, Writing – review and editing; Dominik Haddad, Data curation, Formal analysis, Supervision, Funding acquisition, Investigation, Methodology, Writing – review and editing; Aphroditi Mamaligas, Zac Chatterton, Data curation, Formal analysis, Investigation, Visualization, Methodology, Writing – review and editing; Szu-Chi Liao, Kohei Kano, Data curation, Formal analysis, Investigation, Visualization; Rose Creed, Investigation, Writing – review and editing; Yuhong

Fu, Data curation, Formal analysis, Supervision, Writing – review and editing; Joseph H Garcia, Victoria M Vance, Formal analysis, Investigation; Yoshitaka J Sei, Formal analysis, Methodology; Anatol Kreitzer, Supervision, Methodology; Glenda Halliday, Supervision, Funding acquisition, Visualization, Methodology, Writing – original draft, Project administration, Writing – review and editing; Alexandra B Nelson, Supervision, Funding acquisition, Writing – review and editing; Elyssa Margolis, Data curation, Formal analysis, Funding acquisition, Investigation, Visualization, Methodology, Writing – original draft, Project administration, Writing – review and editing; Ken Nakamura, Conceptualization, Data curation, Supervision, Funding acquisition, Visualization, Methodology, Writing – original draft, Project administration, Writing – review and editing

### Author ORCIDs
Katerina Rademacher ⓘ https://orcid.org/0000-0001-9676-6673
Zak Doric ⓘ https://orcid.org/0000-0003-2608-4196
Dominik Haddad ⓘ https://orcid.org/0009-0001-9354-7747
Aphroditi Mamaligas ⓘ https://orcid.org/0000-0002-9733-6285
Szu-Chi Liao ⓘ https://orcid.org/0000-0003-2744-363X
Rose Creed ⓘ https://orcid.org/0000-0003-4893-2010
Kohei Kano ⓘ https://orcid.org/0000-0002-0223-5781
Zac Chatterton ⓘ https://orcid.org/0000-0002-6683-1400
Yuhong Fu ⓘ https://orcid.org/0000-0003-4539-2039
Joseph H Garcia ⓘ https://orcid.org/0000-0002-7127-9534
Victoria M Vance ⓘ https://orcid.org/0000-0003-4546-5445
Yoshitaka J Sei ⓘ https://orcid.org/0000-0002-4725-8725
Anatol Kreitzer ⓘ https://orcid.org/0000-0001-7423-2398
Glenda Halliday ⓘ https://orcid.org/0000-0003-0422-8398
Alexandra B Nelson ⓘ https://orcid.org/0000-0002-9305-5662
Elyssa Margolis ⓘ https://orcid.org/0000-0001-8777-302X
Ken Nakamura ⓘ https://orcid.org/0000-0002-9192-182X

### Ethics
Human subjects: Midbrain sections were collected from individuals with pathologically verified early-stage Parkinson's Disease (ePD) (n=10) and individuals without any neurological or neuropathological conditions (n=10) through the New South Wales (NSW) Brain Banks, as detailed in Table 2. The research protocol received ethical clearance from the University of Sydney Human Research Ethics Committee (approval number 2021/845).

All animal experimental procedures were conducted in accordance with the Guide for the Care and Use of Laboratory Animals, as adopted by the National Institutes of Health, and with approval from the University of California, San Francisco Institutional Animal Care and Use Committee. All mice were housed in a state-of-the-art barrier facility managed by the UCSF Laboratory Animal Resource Center (LARC). Animal care and use in this research are covered under the UCSF "Assurance of Compliance with PHS Policy on Humane Care and Use of Laboratory Animals by Awardee Institutions" number A3400-01.

Reviewer #1 (Public review): https://doi.org/10.7554/eLife.98775.3.sa1
Reviewer #2 (Public review): https://doi.org/10.7554/eLife.98775.3.sa2
Reviewer #3 (Public review): https://doi.org/10.7554/eLife.98775.3.sa3
Author response https://doi.org/10.7554/eLife.98775.3.sa4

## Additional files

### Supplementary files
MDAR checklist

### Data availability
Data are available at https://doi.org/10.5281/zenodo.10499026. Human spatial transcriptomics data are available at https://zenodo.org/doi/10.5281/zenodo.10499186. The mouse spatial transcriptomics data discussed in this publication have been deposited in NCBI's Gene Expression Omnibus (*Edgar*

*et al., 2002*) and are accessible through GEO Series accession number GSE299398 (https://www.ncbi.nlm.nih.gov/geo/query/acc.cgi?acc=GSE299398). All other data are included in the manuscript and/or supporting information.

The following datasets were generated:

| Author(s) | Year | Dataset title | Dataset URL | Database and Identifier |
|---|---|---|---|---|
| Rademacher K, Doric Z, Sei YJ, Nakamura K | 2025 | Chronic hyperactivation of midbrain dopamine neurons causes preferential dopamine neuron degeneration | https://www.ncbi.nlm.nih.gov/geo/query/acc.cgi?acc=GSE299398 | NCBI Gene Expression Omnibus, GSE299398 |
| Chatterton Z, Fu Y, Halliday G | 2024 | Spatial Transcriptomic analysis (GeoMx) of SNc ventral tier dopamine cells in control and PD subjects | https://doi.org/10.5281/zenodo.10511809 | Zenodo, 10.5281/zenodo.10511809 |
| Rademacher K, Nakamura K, Kreitzer A, Garcia J, Creed R, Nelson A, Doric Z, Haddad D, Mamaligas A, Liao S-C, Kano K, Chatterton Z, Fu Y, Sei Y, Vance V, Halliday G, Margolis E | 2024 | Data for Chronic hyperactivation of midbrain dopamine neurons causes preferential dopamine neuron degeneration | https://doi.org/10.5281/zenodo.15312580 | Zenodo, 10.5281/zenodo.15312580 |

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
