## [Editor Report · eLife Assessment]

This manuscript shows that chronic chemogenetic excitation of dopaminergic neurons in the mouse midbrain results in differential degeneration of axons and somas across distinct regions (SNc vs VTA). These findings are **important** for two reasons. This approach can be used as a mouse model for Parkinson's Disease without the need for the infusion of toxins (e.g. 6-OHDA or MPTP) — this mouse model also has the advantage of showing axon-first degeneration over a time course (2–4 weeks) that is suitable for experimental investigation. Also, the findings that direct excitation of dopaminergic neurons causes differential degeneration sheds light on the mechanisms of dopaminergic neuron selective vulnerability. The evidence that activation of dopaminergic neurons causes degeneration, alters motor behavior, and alters mRNA expression is **convincing**. This is an exciting paper that will have an impact on the Parkinson's Disease field.

---

## [Referee Report · Reviewer #1 (Public review)]

Summary:

In this manuscript, the authors investigated the effect of chronic activation of dopamine neurons using chemogenetics. Using Gq-DREADDs, the authors chronically activated midbrain dopamine neurons and observed that these neurons, particularly their axons, exhibit increased vulnerability and degeneration, resembling the pathological symptoms of Parkinson's disease. Baseline calcium levels in midbrain dopamine neurons were also significantly elevated following the chronic activation. Lastly, to identify cellular and circuit-level changes in response to dopaminergic neuronal degeneration caused by chronic activation, the authors employed spatial genomics (Visium) and revealed comprehensive changes in gene expression in the mouse model subjected to chronic activation. In conclusion, this study presents novel data on the consequences of chronic hyperactivation of midbrain dopamine neurons.

Strengths:

This study provides direct evidence that the chronic activation of dopamine neurons is toxic and gives rise to neurodegeneration. In addition, the authors achieved the chronic activation of dopamine neurons using water application of clozapine-N-oxide (CNO), a method not commonly employed by researchers. This approach may offer new insights into pathophysiological alterations of dopamine neurons in Parkinson's disease. The authors also utilized state-of-the-art spatial gene expression analysis, which can provide valuable information for other researchers studying dopamine neurons. They also presented a substantial number of intriguing ideas in their discussion, which are worth further investigation.

Weaknesses:

Although not fully supported by data, the authors provided a well-explained rationale and proposed possible mechanisms for dopamine neuron degeneration due to chronic activation in their results and discussion.

Comments on revised version:

The authors have adequately addressed most of my comments, and I have no further concerns.

---

## [Referee Report · Reviewer #2 (Public review)]

Rademacher et al. present a paper showing that chronic chemogenetic excitation of dopaminergic neurons in the mouse midbrain results in differential degeneration of axons and somas across distinct regions (SNc vs VTA). These findings are important for two reasons: 1. This approach can be used as a mouse model for Parkinson's Disease without the need for the infusion of toxins (e.g. 6-OHDA or MPTP). This mouse model also has the advantage of showing a axon-first degeneration over an experimentally-useful time course (2-4 weeks). 2. The findings that direct excitation of dopaminergic neurons causes differential degeneration sheds light on the mechanisms of dopaminergic neuron selective vulnerability. The evidence that activation of dopaminergic neurons causes degeneration, alters motor behavior, and alters mRNA expression is convincing. This is an exciting and important paper and will have an impact on the Parkinson's Disease field.

Strengths:

This is an exciting and important paper and will have an impact on the Parkinson's Disease field.

It presents a new highly useful mouse model of PD.

The paper compares mouse transcriptomics with human patient data.

It shows that selective degeneration can occur across the midbrain dopaminergic neurons even in the absence of a genetic, prion, or toxin neurodegeneration mechanism.

Weaknesses:

The authors have addressed all my concerns. This is an interesting, important, and carefully-controlled study.

---

## [Referee Report · Reviewer #3 (Public review)]

Summary:

In this manuscript, Rademacher and colleagues examined the effect on the integrity of the dopamine system in mice of chronically stimulating dopamine neurons using a chemogenetic approach. They find that one to two weeks of constant exposure to the chemogenetic activator CNO leads to a decrease in the density of tyrosine hydroxylase staining in striatal brain sections and to a small reduction of the global population of tyrosine hydroxylase positive neurons in the ventral midbrain. They also report alterations in gene expression in both regions using a spatial transcriptomics approach. Globally, the work is well done and valuable and some of the conclusions are interesting. However, the conceptual advance is perhaps a bit limited in the sense that there is extensive previous work in the literature showing that excessive depolarization of multiple types of neurons associated with intracellular calcium elevations promotes neuronal degeneration. The present work adds to this by showing evidence of a similar phenomenon in dopamine neurons. In terms of the mechanisms explaining the neuronal loss observed after 2 to 4 weeks of chemogenetic activation, it would be important to consider that dopamine neurons are known from a lot of previous literature to undergo a decrease in firing through a depolarization-block mechanism when chronically depolarized. Is it possible that such a phenomenon explains much of the results observed in the present study? It would be important to consider this in the manuscript. The relevance to Parkinson's disease (PD) is also not totally clear because there is not a lot of previous solid evidence showing that the firing of dopamine neurons is increased in PD, either in human subjects or in mouse models of the disease.

Comments on revisions:

The authors have done a good job at revising the manuscript. The revised manuscript better frames the results in the context of previous literature.

---

## [Author Response]

The following is the authors’ response to the original reviews

**Reviewer #1 (Public Review):**
Summary:In this manuscript, the authors investigated the effect of chronic activation of dopamine neurons using chemogenetics. Using Gq-DREADDs, the authors chronically activated midbrain dopamine neurons and observed that these neurons, particularly their axons, exhibit increased vulnerability and degeneration, resembling the pathological symptoms of Parkinson's disease. Baseline calcium levels in midbrain dopamine neurons were also significantly elevated following the chronic activation. Lastly, to identify cellular and circuit-level changes in response to dopaminergic neuronal degeneration caused by chronic activation, the authors employed spatial genomics (Visium) and revealed comprehensive changes in gene expression in the mouse model subjected to chronic activation. In conclusion, this study presents novel data on the consequences of chronic hyperactivation of midbrain dopamine neurons.Strengths:This study provides direct evidence that the chronic activation of dopamine neurons is toxic and gives rise to neurodegeneration. In addition, the authors achieved the chronic activation of dopamine neurons using water application of clozapine-N-oxide (CNO), a method not commonly employed by researchers. This approach may offer new insights into pathophysiological alterations of dopamine neurons in Parkinson's disease. The authors also utilized state-of-the-art spatial gene expression analysis, which can provide valuable information for other researchers studying dopamine neurons. Although the authors did not elucidate the mechanisms underlying dopaminergic neuronal and axonal death, they presented a substantial number of intriguing ideas in their discussion, which are worth further investigation.

We thank the reviewer for these positive comments.

Weaknesses:Many claims raised in this paper are only partially supported by the experimental results. So, additional data are necessary to strengthen the claims. The effects of chronic activation of dopamine neurons are intriguing; however, this paper does not go beyond reporting phenomena. It lacks a comprehensive explanation for the degeneration of dopamine neurons and their axons. While the authors proposed possible mechanisms for the degeneration in their discussion, such as differentially expressed genes, these remain experimentally unexplored.

We thank the reviewer for this review. We do believe that the manuscript has a substantial mechanistic component, as the central experiments involve direct manipulation of neuronal activity, and we show an increase in calcium levels and gene expression changes in dopamine neurons that coincide with the degeneration. However, we agree that deeper mechanistic investigation would strengthen the conclusions of the paper. We have executed several important revisions, including the addition of CNO behavioral controls, manipulation of intracellular calcium using isradipine, additional transcriptomics experiments and further validation of findings. We believe that these additions significantly bolster the conclusions of the paper.

**Reviewer #2 (Public Review):**
Summary:Rademacher et al. present a paper showing that chronic chemogenetic excitation of dopaminergic neurons in the mouse midbrain results in differential degeneration of axons and somas across distinct regions (SNc vs VTA). These findings are important. This mouse model also has the advantage of showing a axon-first degeneration over an experimentally-useful time course (2-4 weeks). 2. The findings that direct excitation of dopaminergic neurons causes differential degeneration sheds light on the mechanisms of dopaminergic neuron selective vulnerability. The evidence that activation of dopaminergic neurons causes degeneration and alters mRNA expression is convincing, as the authors use both vehicle and CNO control groups, but the evidence that chronic dopaminergic activation alters circadian rhythm and motor behavior is incomplete as the authors did not run a CNO-control condition in these experiments.Strengths:This is an exciting and important paper.The paper compares mouse transcriptomics with human patient data.It shows that selective degeneration can occur across the midbrain dopaminergic neurons even in the absence of a genetic, prion, or toxin neurodegeneration mechanism.

We thank the reviewer for these comments.

Weaknesses:Major concerns:(1) The lack of a CNO-positive, DREADD-negative control group in the behavioral experiments is the main limitation in interpreting the behavioral data. Without knowing whether CNO on its own has an impact on circadian rhythm or motor activity, the certainty that dopaminergic hyperactivity is causing these effects is lacking.

We thank the reviewer for this important recommendation. Although the initial version showed that CNO does not produce degeneration of DA neuron terminals, it did not exclude a contribution to the behavioral changes. To address this, we now include a cohort of DREADD free non-injected mice treated with either vehicle or CNO (Figure S1C). We found that on its own, CNO did not significantly impact either light cycle or dark cycle running. Together these results along with the lack of degeneration observed with CNO treatment in non-DREADD mice (Figure 2D) support that our behavioral and histological results are the result of dopamine neuron activation.

(2) One of the most exciting things about this paper is that the SNc degenerates more strongly than the VTA when both regions are, in theory, excited to the same extent. However, it is not perfectly clear that both regions respond to CNO to the same extent. The electrophysiological data showing CNO responsiveness is only conducted in the SNc. If the VTA response is significantly reduced vs the SNc response, then the selectivity of the SNc degeneration could just be because the SNc was more hyperactive than the VTA. Electrophysiology experiments comparing the VTA and SNc response to CNO could support the idea that the SNc has substantial intrinsic vulnerability factors compared to the VTA.

We agree that additional electrophysiology conducted in the VTA dopamine neurons would meaningfully add to our understanding of the selective vulnerability in this model, and have completed these experiments in the revision (Figure 1, Figure S2). We now show that in vivo treatment with CNO causes some of the same physiological changes in VTA dopamine neurons as we found in SNc dopamine neurons, including an increased spontaneous firing rate, and a similar decrease in responsiveness to CNO in the slice recordings. Together these observations support the conclusion that SNc axons are intrinsically more vulnerable to increased activity than VTA dopamine axons.

(3) The mice have access to a running wheel for the circadian rhythm experiments. Running has been shown to alter the dopaminergic system (Bastioli et al., 2022) and so the authors should clarify whether the histology, electrophysiology, fiber photometry, and transcriptomics data are conducted on mice that have been running or sedentary.

We have clarified which mice had access to a running wheel in the methods of our revision. Briefly, mice for histology, electrophysiology, and transcriptomics all had access to a running wheel during their treatment. The mice used for photometry underwent about 7 days of running wheel access approximately 3 weeks prior to the beginning of the experiment. The photometry headcaps prevented mice from having access to a running wheel in their home cage. Mice used for non-responder and non-hM3Dq (CNO alone) experiments also had access to a running wheel during their treatment. Mice used for the isradipine experiment did not have access to a running wheel, as the number of mice was too large and while unilateral hM3Dq expression allows for within-animal controls, it does not lend to clear interpretation of running wheel data.

**Reviewer #3 (Public Review):**
Summary:In this manuscript, Rademacher and colleagues examined the effect on the integrity of the dopamine system in mice of chronically stimulating dopamine neurons using a chemogenetic approach. They find that one to two weeks of constant exposure to the chemogenetic activator CNO leads to a decrease in the density of tyrosine hydroxylase staining in striatal brain sections and to a small reduction of the global population of tyrosine hydroxylase positive neurons in the ventral midbrain. They also report alterations in gene expression in both regions using a spatial transcriptomics approach. Globally, the work is well done and valuable and some of the conclusions are interesting. However, the conceptual advance is perhaps a bit limited in the sense that there is extensive previous work in the literature showing that excessive depolarization of multiple types of neurons associated with intracellular calcium elevations promotes neuronal degeneration. The present work adds to this by showing evidence of a similar phenomenon in dopamine neurons.

We thank the reviewer for the careful and thoughtful review of our manuscript.

While extensive depolarization and associated intracellular calcium elevations promote degeneration generally, we emphasize that the process we describe is novel. Indeed, prior studies delivering chronic DREADDs to vulnerable neurons in models of Alzheimer’s disease did not detect an increase in neurodegeneration, despite seeing changes in protein aggregation (e.g. Yuan and Grutzendler, J Neurosci 2016, PMID: 26758850; Hussaini et al., PLOS Bio 2020, PMID: 32822389). Further, a critical finding from our study is that in our paradigm, this stressor does not impact all dopamine neurons equally, as the SNc DA neurons are more vulnerable than VTA DA neurons, mirroring selective vulnerability characteristic of Parkinson’s disease. This is consistent with a large body of literature that SNc dopamine neurons are less capable of handling large energetic and calcium loads compared to neighboring VTA neurons, and the finding that chronically altered activity is sufficient to drive this preferential loss is novel. In addition, we are not aware of prior studies that have chronically activated DREADDs over several weeks to produce neurodegeneration.

In terms of the mechanisms explaining the neuronal loss observed after 2 to 4 weeks of chemogenetic activation, it would be important to consider that dopamine neurons are known from a lot of previous literature to undergo a decrease in firing through a depolarization-block mechanism when chronically depolarized. Is it possible that such a phenomenon explains much of the results observed in the present study? It would be important to consider this in the manuscript.

Thank you for this comment. As discussed in greater detail in the “comments on results section” below, our data suggests this isn’t a prominent feature in our model. However, we cannot rule out a contribution of depolarization block, and have expanded on the discussion of this possibility in the revised manuscript.

The relevance to Parkinson's disease (PD) is also not totally clear because there is not a lot of previous solid evidence showing that the firing of dopamine neurons is increased in PD, either in human subjects or in mouse models of the disease. As such, it is not clear if the present work is really modelling something that could happen in PD in humans.

We completely agree that evidence of increased dopamine neuron activity from human PD patients is lacking, and the little data that exists is difficult to interpret without human controls. However, as we outline in the manuscript, multiple lines of evidence suggest that the activity level of dopamine neurons almost certainly does change in PD. Therefore, it is very important that we understand how changes in the level of neural activity influence the degeneration of DA neurons. In this paper we examine the impact of increased activity. Increased activity may be compensatory after initial dopamine neuron loss, or may be an initial driver of death (Rademacher & Nakamura, Exp Neurol 2024, PMID: 38092187). In addition to the human and rodent data already discussed in the manuscript, additional support for increased activity in PD models include:

• Elevated firing rates in asymptomatic MitoPark mice (Good et al., FASEB J 2011, PMID: 21233488)

• Increased frequency of spontaneous firing in patient-derived iPSC dopamine neurons and primary mouse dopamine neurons that overexpress synuclein (Lin et al., Acta Neuropath Comm 2021, PMID: 34099060)

• Increased spontaneous firing in dopamine neurons of rats injected with synuclein preformed fibrils compared to sham (Tozzi et al., Brain 2021, PMID: 34297092)

We have included citation of these important examples in our revision. In our model, we have found that chronic hyperactivity causes a substantial loss of nigral DA terminals while mesolimbic terminals are relatively spared (Figure 2), and that striatal DA levels are markedly decreased (Figure S6), phenomena that are hallmarks of Parkinson’s disease.

There are additional levels of complexity to accurately model changes in PD, which may differ between subtypes of the disease, the disease stage, and the subtype of dopamine neuron. Our study models a form of increased intrinsic activity, and interpretation of our results will be facilitated as we learn more about how the activity of DA neurons changes in humans in PD. Similarly, in future studies, it will also be important to study the impact of decreasing DA neuron activity.

Comments on the introduction:The introduction cites a 1990 paper from the lab of Anthony Grace as support of the fact that DA neurons increase their firing rate in PD models. However, in this 1990 paper, the authors stated that: "With respect to DA cell activity, depletions of up to 96% of striatal DA did not result in substantial alterations in the proportion of DA neurons active, their mean firing rate, or their firing pattern. Increases in these parameters only occurred when striatal DA depletions exceeded 96%." Such results argue that an increase in firing rate is most likely to be a consequence of the almost complete loss of dopamine neurons rather than an initial driver of neuronal loss. The present introduction would thus benefit from being revised to clarify the overriding hypothesis and rationale in relation to PD and better represent the findings of the paper by Hollerman and Grace.

We agree that the findings of Hollerman and Grace support compensatory changes in dopamine neuron activity in response to loss of dopamine neurons, rather than informing whether dopamine neuron loss can also be an initial driver of activity. Importantly, while significant changes to burst firing were not seen until almost complete loss of dopamine neurons, these recordings were made in anesthetized rats which may not be representative of neural activity in awake animals. We adjusted the text so that this is no longer referred to as ‘partial’ loss. At the same time, we point out that the results of other studies on this point are mixed: a 50% reduction in dopamine neurons didn’t alter firing rate or bursting (Harden and Grace, J Neurosci 1995, PMID: 7666198; Bilbao et al., Brain Res 2006, PMID: 16574080), while a 40% loss was found to increase firing rate and bursting (Chen et al., Brain Res 2009. PMID: 19545547) and larger reductions alter burst firing (Hollerman & Grace, Brain Res 1990, PMID: 2126975; Stachowiak et al., J Neurosci 1987, PMID: 3110381). Importantly, even if compensatory, such late-stage increases in dopamine neuron activity may contribute to disease progression and drive a vicious cycle of degeneration in surviving neurons. In addition, we also don’t know how the threshold of dopamine neuron loss and altered activity may differ between mice and humans, and PD patients do not present with clinical symptoms until ~30-60% of nigral neurons are lost (Burke & O’Malley, Exp Neurol 2013, PMID: 22285449; Shulman et al., Annu Rev Pathol 2011, PMID: 21034221).

Other lines of evidence support the potential role of hyperactivity in disease initiation, including increased activity before dopamine neuron loss in MitoPark mice (Good et al., FASEB J 2011, PMID: 21233488), increased spontaneous firing in patient-derived iPSC dopamine neurons (Lin et al., Acta Neuropath Comm 2021, PMID: 34099060), and increased activity observed in genetic models of PD (Bishop et al., J Neurophysiol 2010, PMID: 20926611; Regoni et al., Cell Death Dis 2020, PMID: 33173027).

It would be good that the introduction refers to some of the literature on the links between excessive neuronal activity, calcium, and neurodegeneration. There is a large literature on this and referring to it would help frame the work and its novelty in a broader context.

We agree that a discussion of hyperactivity, calcium, and neurodegeneration would benefit the introduction. Accordingly, we have expanded on our citation of this literature in both the introduction and discussion sections. However, we believe that the novelty of our study lies in: (1) a chronic chemogenetic activation paradigm via drinking water, (2) demonstrating selective vulnerability of dopamine neurons as a result of altering their activity/excitability alone, and (3) comparing mouse and human spatial transcriptomics.

Comments on the results section:The running wheel results of Figure 1 suggest that the CNO treatment caused a brief increase in running on the first day after which there was a strong decrease during the subsequent days in the active phase. This observation is also in line with the appearance of a depolarization block.The authors examined many basic electrophysiological parameters of recorded dopamine neurons in acute brain slices. However, it is surprising that they did not report the resting membrane potential, or the input resistance. It would be important that this be added because these two parameters provide key information on the basal excitability of the recorded neurons. They would also allow us to obtain insight into the possibility that the neurons are chronically depolarized and thus in depolarization block.

We do report the input resistance in Figure S1C (now Figure S2A, S2B), which was unchanged in CNO-treated animals compared to controls. We did not previously report the resting membrane potential because many of the DA neurons were spontaneously firing. In the revision, we now report the initial membrane potential on first breaking into the cell for the whole cell recordings, which did not vary between groups (Figure S2). This is still influenced by action potential activity, but is the timepoint in the recording least impacted by dialyzing the neuron with the internal solution, which might alter the intracellular concentrations of ions. We observed increased spontaneous action potential activity ex vivo in slices from CNO-treated mice (Figure 1D), thus at least under these conditions these dopamine neurons are not in depolarization block. We also did not see strong evidence of changes in other intrinsic properties of the neurons with whole cell recordings (e.g. Figure S2). Overall, our electrophysiology experiments are not consistent with the depolarization block model, at least not due to changes in the intrinsic properties of the neurons. Although our ex vivo findings cannot exclude a contribution of depolarization block in vivo, we do show that CNO-treated mice removed from their cages for open field testing continue to have a strong trend for increased activity for approximately 10 days (Figure S4B). This finding is also consistent with increased activity of the DA neurons. We have added discussion of these important considerations in the revision.

It is great that the authors quantified not only TH levels but also the levels of mCherry, coexpressed with the chemogenetic receptor. This could in principle help to distinguish between TH downregulation and true loss of dopamine neuron cell bodies. However, the approach used here has a major caveat in that the number of mCherry-positive dopamine neurons depends on the proportion of dopamine neurons that were infected and expressed the DREADD and this could very well vary between different mice. It is very unlikely that the virus injection allowed to infect 100% of the neurons in the VTA and SNc. This could for example explain in part the mismatch between the number of VTA dopamine neurons counted in panel 2G when comparing TH and mCherry counts. Also, I see that the mCherry counts were not provided at the 2-week time point. If the mCherry had been expressed genetically by crossing the DAT-Cre mice with a floxed fluorescent reported mice, the interpretation would have been simpler. In this context, I am not convinced of the benefit of the mCherry quantifications. The authors should consider either removing these results from the final manuscript or discussing this important limitation.

We thank the reviewer for this comment, and we agree that this is a caveat of our mCherry quantification. Quantitation of the number of mCherry+ DA neurons specifically informs the impact on transduced DA neurons, and mCherry appears to be less susceptible to downregulation versus TH. As the reviewer points out, it carries the caveat that there is some variability between injections. Our control animals give us an indicator of injection variability, which is likely substantial and prevents us from detecting more subtle changes. Nonetheless, we believe that it conveys useful complementary data. We discuss this caveat in our revision. Note that mCherry was not quantified at the two-week timepoint because there is no loss of TH+ cells at that time.

Although the authors conclude that there is a global decrease in the number of dopamine neurons after 4 weeks of CNO treatment, the post-hoc tests failed to confirm that the decrease in dopamine number was significant in the SNc, the region most relevant to Parkinson's. This could be due to the fact that only a small number of mice were tested. A "n" of just 4 or 5 mice is very small for a stereological counting experiment. As such, this experiment was clearly underpowered at the statistical level. Also, the choice of the image used to illustrate this in panel 2G should be reconsidered: the image suggests that a very large loss of dopamineneurons occurred in the SNc and this is not what the numbers show. A more representative image should be used.

We agree that the stereology experiments were performed on relatively small numbers of animals, such that only robust effects would be detected. Combined with the small effect size, this may have contributed to the post-hoc tests showing a trend of p=0.1 for both the TH and mCherry dopamine cell counts in the SN at 4 weeks. Given this small effect size, we would indeed need much larger groups to better discern these changes. Stereology is an intensive technique, and we have therefore elected to focus on terminal loss. We have also replaced panel 2G with a more representative CNO image.

In Figure 3, the authors attempt to compare intracellular calcium levels in dopamine neurons using GCaMP6 fluorescence. Because this calcium indicator is not quantitative (unlike ratiometric sensors such as Fura2), it is usually used to quantify relative changes in intracellular calcium. The present use of this probe to compare absolute values is unusual and the validity of this approach is unclear. This limitation needs to be discussed. The authors also need to refer in the text to the difference between panels D and E of this figure. It is surprising that the fluctuations in calcium levels were not quantified. I guess the hypothesis was that there should be more or larger fluctuations in the mice treated with CNO if the CNO treatment led to increased firing. This needs to be clarified.

We thank the reviewer for this comment. We understand that this method of comparing absolute values is unconventional. However, these animals were tested concurrently on the same system, and a clear effect on the absolute baseline was observed. We have included a caveat of this in our discussion. Panel D of this figure shows the raw, uncorrected photometry traces, whereas panel E shows the isosbestic corrected traces for the same recording. In panel E, the traces follow time in ascending order. We have also included frequency and amplitude data for these recordings (Figure S4A), along with discussion of the significance of these findings.

Although the spatial transcriptomic results are intriguing and certainly a great way to start thinking about how the CNO treatment could lead to the loss of dopamine neurons, the presented results, the focusing of some broad classes of differentially expressed genes and on some specific examples, do not really suggest any clear mechanism of neurodegeneration. It would perhaps be useful for the authors to use the obtained data to validate that a state of chronic depolarization was indeed induced by the chronic CNO treatment. Were genes classically linked to increased activity like cfos or bdnf elevated in the SNc or VTA dopamine neurons? In the striatum, the authors report that the levels of DARP32, a gene whose levels are linked to dopamine levels, are unchanged. Does this mean that there were no major changes in dopamine levels in the striatum of these mice?

While levels of DARPP32 mRNA were unchanged, our additional HPLC data show strong decreases in striatal dopamine in hyperactivated mice. We do not see strong changes in classic activity-related genes (data not shown), however these genes may behave differently in the context of chronic hyperactivity and ongoing degeneration. Instead, we employed NEUROeSTIMator (Bahl et al., Nature Comm. 2024, PMID: 38278804), a deep learning method to predict neural activation based on transcriptomic data. We found that predicted activity scores were significantly higher in GqCNO dopaminergic regions compared to controls (Figure X). Indeed, some of the genes used within the model to predict activity are immediate early genes eg. c-fos.

The usefulness of comparing the transcriptome of human PD SNc or VTA sections to that of the present mouse model should be better explained. In the human tissues, the transcriptome reflects the state of the tissue many years after extensive loss of dopamine neurons. It is expected that there will be few if any SNc neurons left in such sections. In comparison, the mice after 7 days of CNO treatment do not appear to have lost any dopamine neurons. As such, how can the two extremely different conditions be reasonably compared? Our mouse model and human PD progress over distinct timescales, as is the case with essentially all mouse models of neurodegenerative diseases. Nonetheless, in our view there is still great value in comparing gene expression changes in mouse models with those in human disease. It seems very likely that the same pathologic processes that drive degeneration early in the disease continue to drive degeneration later in the disease. Note that we have tried to address the discrepancy in time scales in part by comparing our mouse model to early PD samples when there is more limited SNc DA neuron loss (see the proportion of DA neurons within the areas of human tissues we selected for sampling in Author response image 1). Therefore, we can indeed use spatial transcriptomics to compare dopamine neurons from mice with initial degeneration to those in patients where degeneration is ongoing.

**Author response image 1. sa4fig1:** Violin plot of DA neuron proportions sampled within the vulnerable SNV (deconvoluted RCTD method used in unmasked tissue sections of the SNV). Control and early PD subjects.

Comments on the discussion:In the discussion, the authors state that their calcium photometry results support a central role of calcium in activity-induced neurodegeneration. This conclusion, although plausible because of the very broad pre-existing literature linking calcium elevation (such as in excitotoxicity) to neuronal loss, should be toned down a bit as no causal relationship was established in the experiments that were carried out in the present study.

Our model utilizes hM3Dq-DREADDs that function by activating Gq pathways that are classically expected to increase intracellular calcium to increase neuronal excitability. Indeed in slices from mice that were not treated with CNO, acute CNO application caused depolarizations (Figure 1E) that can be due to an increase in intracellular calcium and also cause increases in intracellular calcium. Additionally, our results show increased calcium by fiber photometry and changes to calcium-related genes, suggesting a causal relation and crucial role of calcium in the mechanism of degeneration. However, we agree that we have not experimentally proven this point. Indeed, a small preliminary experiment with chronic isradipine failed to show protection, although it lacked power to detect a partial effect. We have acknowledged this in the text, and also briefly consider other mechanisms such as increased dopamine levels that could also mediate the toxicity.

In the discussion, the authors discuss some of the parallel changes in gene expression detected in the mouse model and in the human tissues. Because few if any dopamine neurons are expected to remain in the SNc of the human tissues used, this sort of comparison has important conceptual limitations and these need to be clearly addressed.

As discussed, we sampled SN DA neurons in early PD (see Author response image 1), and in our view there is great value for such comparisons.

A major limitation of the present discussion is that it does not discuss the possibility that the observed phenotypes are caused by the induction of a chronic state of depolarization block by the chronic CNO treatment. I encourage the authors to consider and discuss this hypothesis.

As discussed above, our analyses of DA neuron firing in slices and open field testing to date do not support a prominent contribution of depolarization block with chronic CNO treatment. However, we cannot rule out this hypothesis, therefore we have included additional electrophysiology experiments and have added discussion of this important consideration.

Also, the authors need to discuss the fact that previous work was only able to detect an increase in the firing rate of dopamine neurons after more than 95% loss of dopamine neurons. As such, the authors need to clearly discuss the relevance of the present model to PD. Are changes in firing rate a driver of neuronal loss in PD, as the authors try to make the case here, or are such changes only a secondary consequence of extensive neuronal loss (for example because a major loss of dopamine would lead to reduced D2 autoreceptor activation in the remaining neurons, and to reduced autoreceptor-mediated negative feedback on firing). This needs to be discussed.

As discussed above, while increases in dopamine neuron activity may be compensatory after loss of neurons, the precise percentage required to induce such compensatory changes is not defined in mice and varies between paradigms, and the threshold level is not known in humans. We also reiterate that a compensatory increase in activity could still promote the degeneration of critical surviving DA neurons, whose loss underlies the substantial decline in motor function that typically occurs over the course of PD. Moreover, there are also multiple lines of evidence to suggest that changes in activity can initiate and drive dopamine neuron degeneration (Rademacher & Nakamura, Exp Neurol 2024). For example, overexpression of synuclein can increase firing in cultured dopamine neurons (Dagra et al., NPJ Parkinsons Dis 2021, PMID: 34408150), while mice expressing mutant Parkin have higher mean firing rates (Regoni et al., Cell Death Dis 2020, PMID: 33173027). Similarly, an increased firing rate has been reported in the MitoPark mouse model of PD at a time preceding DA neuron degeneration (Good et al., FASEB J 2011, PMID: 21233488). We also acknowledge that alterations to dopamine neuron activity are likely complex in PD, and that dopamine neuron health and function can be impacted not just by simple increases in activity, but also by changes in activity patterns and regularity. We have amended our discussion to include the important caveat of changes in activity occurring as compensation, as well as further evidence of changes in activity preceding dopamine neuron death.

There is a very large, multi-decade literature on calcium elevation and its effects on neuronal loss in many different types of neurons. The authors should discuss their findings in this context and refer to some of this previous work. In a nutshell, the observations of the present manuscript could be summarized by stating that the chronic membrane depolarization induced by the CNO treatment is likely to induce a chronic elevation of intracellular calcium and this is then likely to activate some of the well-known calcium-dependent cell death mechanisms. Whether such cell death is linked in any way to PD is not really demonstrated by the present results. The authors are encouraged to perform a thorough revision of the discussion to address all of these issues, discuss the major limitations of the present model, and refer to the broad pre-existing literature linking membrane depolarization, calcium, and neuronal loss in many neuronal cell types.

While our model demonstrates classic excitotoxic cell death pathways, we would like to emphasize both the chronic nature of our manipulation and the progressive changes observed, with increasing degeneration seen at 1, 2, and 4 weeks of hyperactivity in an axon-first manner. This is a unique aspect of our study, in contrast to much of the previous literature which has focused on shorter timescales. Thus, while we have revised the discussion to more comprehensively acknowledge previous studies of calcium-dependent neuron cell death, we believe we have made several new contributions that are not predicted by existing literature. We have shown that this chronic manipulation is specifically toxic to nigral dopamine neurons, and the data that VTA dopamine neurons continue to be resilient even at 4 weeks is interesting and disease-relevant. We therefore do not want to use findings from other neuron types to draw assumptions about DA neurons, which are a unique and very diverse population. We acknowledge that as with all preclinical models of PD, we cannot draw definitive conclusions about PD with this data. However, we reiterate that we strongly believe that drawing connections to human disease is important, as dopamine neuron activity is very likely altered in PD and a clearer understanding of how dopamine neuron survival is impacted by activity will provide insight into the mechanisms of PD.

**Recommendations for the authors:**

**Reviewer #1 (Recommendations For The Authors):**
(1) The temporal design of the experiments is quite confusing. For instance, Figures 1 and 3 illustrate the daily changes of the mice and suggest some critical time points within 2 weeks of CNO administration, whereas Figure 2 presents data at 2 and 4 weeks, which are much later than the proposed critical time points. Furthermore, Figure 4 includes only 1 week data, and lacks subsequent data from 2 and 4 weeks, at which significant changes such as calcium levels and neuronal/axonal degeneration are observed.

While interesting behavior and calcium phenotypes were detected within 2 and 4 weeks of CNO administration (Figures 1 and 3), we only collected tissues for histology at the 2 and 4 week time points (Figure 2). Observing degeneration of DA neuron axons but not cell bodies at 2 weeks served as a rationale to extend to the 4 week time point to determine whether degeneration was progressive. At the same time, our primary focus is on identifying early changes that may drive or contribute to the degeneration. As such, we recorded calcium changes over a 2-week treatment period, capturing the period during which almost all of the dopamine axons are lost. Similarly, we had the capacity to perform spatial transcriptomics at only one time point, and the 1 week time point was selected to capture transcriptomic changes that precede and potentially contribute to the mild and severe degeneration that occurs at 2 and 4 weeks, respectively. We have added text clarifying the rationale for the time points chosen.

(2) The authors showed the changes in neuronal firing in dopamine neurons by the administration of CNO. However, one of the most important features of dopaminergic neuronal activity is dopamine release at its axon terminals in the striatum. Thus, the claims raised in this paper would be better supported if the authors further show any alterations in dopamine release (by FSCV or fluorescent dopamine sensors) at some critical time points during or after CNO application.

While we are confident that DA release is altered due to the significant changes in behavior when hM3Dq DREADDs are activated specifically in DA neurons, the current manuscript does not quantify this, or distinguish between axonal and somatodendritic DA release. Interestingly, we did find significantly decreased striatal dopamine by HPLC after chronic activation (Figure S6). We believe that resolving these questions is beyond the scope of this manuscript, but have added text indicating the importance of these experiments.

(3) The authors used 2% sucrose as a vehicle via drinking water. Please explain the rationale behind this choice.

We used 2% sucrose as the vehicle because it is also added to the CNO water to counteract the bitterness of CNO (Kumar et al., J Neurotrauma 2024, PMID: 37905504). We have clarified this in the manuscript.

(4) As we know, mRNA levels of some genes do not always predict their protein levels; there is sometimes a huge discrepancy between mRNA and protein abundance. In this paper, the mechanistic interpretation of the results by the authors heavily relies on the spatial transcriptomics of the midbrain and striatum. Thus, the authors need to provide additional data proving that the gene expression of some genes in the CNO group is also changed at the level of protein.

We agree that validating hits at the protein level is valuable, however we were limited in our ability to assess these changes for the revision. However, we have done additional transcriptomics with the high resolution Xenium platform to increase confidence in a subset of hits of interest for follow up in future work, and we included data on genes related to DA metabolism and markers of DA neurons.

(5) The authors provided spatial transcriptomics data only for mice with one week of chronic activation. However, other data also indicate significant differences when the activation period extends beyond 10 to 12 days (Figure 1C, Figure 3D-F). While a 7-day chronic activation time point might be crucial, additional transcriptomics data from later time points would be beneficial to confirm the persistence of these changes in gene expression. Furthermore, differential gene expression (DEG) analysis at these later time points could identify novel pathways or genes influenced by the chronic activation of dopamine neurons.

This is an interesting point and would provide valuable data as to how chronic activity influences gene expression, however additional transcriptomics at later timepoints is beyond the scope of this paper. In future studies we will assess changes observed in this manuscript at other time points.

(6) Figure 1D, Figure S1C:The authors should present the sample recording traces to demonstrate that the electrophysiological recordings were appropriately made.

These data have been provided in Figure S2.

(7) Figure S1C:AP thresholds in SNc dopamine neurons from both groups look quite high. In addition, considering the data from the previous reports, AP peak amplitudes in SNc dopamine neurons from both groups seem to be very low. Are these values correct?

The thresholds and peaks are correct, including the AP (threshold to peak), which is typical in our (Dr. Margolis’s) experience. AP thresholds are measured from an average of at least 10 APs, as the voltage at which the derivative of the trace first exceeds 10 V/s. As mentioned in the methods section, junction potentials were not corrected, which can result in values that are a bit depolarized from ground truth. This junction potential would be consistent across all recordings, thus not impede detection of a difference in AP thresholds between groups of animals.

(8) Figure 1E:It would be better if the statistical significance is depicted in the graph.

We don’t perform repeated measures statistics across data like these, as the data are continuous, collected at 10 kHz. For ease of displaying the data, the data for each neuron is binned and then these traces are averaged together. We display SEM to give a sense of the variance across neurons. We have provided sample traces of individual neurons to better demonstrate the variability and significance of this data (Figure S2).

(9) Figure 2C:The representative staining images appear to be taken from coronal slices at anatomically different positions along the rostral-to-caudal axis. Although the total numbers of TH+ cells are comparable between vehicle and CNO groups in the graph, the sample images do not reflect this result. The authors should replace the current images with the better ones.

We have replaced this image in the manuscript.

**Reviewer #2 (Recommendations For The Authors):**
Minor concerns:(1) The authors claim that their transcriptomics experiments are conducted 'before any degeneration has occurred'. And they do not see significant differences in the TH expression in the striatum. However, the n for these mice at 1 week is lower than the n use at 2 weeks (n=5 vs n=8-9) and the images used to show 'no degeneration' really look like there is some degeneration going on. Also, throughout the paper, there is a stronger effect when degeneration is measured with mCherry compared to when it is measured with TH. The 'no change' claim is made only with the TH comparison. It seems possible (and almost likely) that there would be significant axonal degeneration at one week with either a higher sample size or using the mCherry comparison. The authors should simply claim that their transcriptomics data is collected before any 'somatic' degeneration occurs.

Thank you, we have included data that shows partial terminal loss after one week of activation (Figure S3B, Figure S5A) and have corrected this language in the manuscript to reflect transcriptomics occurring before somatic degeneration.

(2) While selective degeneration is one of the most interesting findings in the paper, that finding is not emphasized and why it would be interesting to compare the VTA vs SNc is not discussed in the introduction.

Emphasis for comparing the VTA vs the SNc has been added to the introduction, along with additional electrophysiology data in VTA dopamine neurons in Figure 1 and Figure S2.

(3) In a similar direction, the vulnerability of dopaminergic neurons has been shown to be differential even within the SNc, with the ventral tier neurons degenerating more severely and the dorsal tier neurons remaining resilient. Is there any evidence for a ventral-dorsal degeneration gradient in the SNc in these experiments?

This is a really interesting point and changes to dopamine neuron subtypes along the ventraldorsal axis may be occurring in this model, particularly as there is more selective loss of SNc neurons. However, the cell type involved would be difficult to determine at this stage, since single cell transcriptomic resolution is necessary across the entire SNc to identify cell subtypes. Transcriptomic identification is further complicated given that transcriptome change has recently been shown with genetic manipulation (Gaertner *et al.*, bioRxiv 2024, PMID: 38895448), and we would think could similarly change with increased activity. Assessing these issues are beyond the scope of this paper.

(4) The running data is very interesting and the circadian rhythm alterations are compelling.However, it is unclear whether the CNO mice run more total compared with the vehicle mice.

The authors should show the combined total running data to evaluate this. We now show total running data in Figure 1C.

(5) The finding that acute CNO has no effect on the membrane potential of SNc neurons after chronic CNO exposure is very peculiar! Especially because the fiber photometry data suggests that CNO continues to have an effect in vivo. Is there any explanation for this?

While there is no acute electrophysiological response to CNO detected in this group, there may be intracellular pathways activated by the DREADD that do not acutely impact membrane potential in current clamp (I = 0 pA) mode.

(6) The terminology of chronic CNO is sometimes confusing as it refers to both 2-week and 4week administration. Using additional terminology such as 'early' and 'late' might help with clarity.

We have decreased usage of ‘chronic,’ and increased usage of more specific treatment times in order to increase clarity throughout the manuscript.

(7) In Figure 2C, the SNc image looks binarized.

This image has been updated.

(8) Also in Figure 2, why are TH and mCherry measured for the 4-week time point, but only TH measured for the 2-week time point?

mCherry quantification was performed to further support the finding of DA neuron death, and was therefore not assessed at 2 weeks given that there was no change in the TH stereology.

(9) Additional scale bars and labeling is needed in Figure 3. In addition, there is such a strong reduction in noise after chronic CNO in the fiber photometry recordings, and the noise does not return upon CNO washout. What is the explanation for this?

Additional scale bars were added to Figure 3. Traces are not getting less noisy with chronic CNO treatment, rather, there is less bursting activity in the dopamine cells. Our interpretation is that the baseline activity is rescued during washout but this bursting activity is not.

(10) While not necessary to support the claims in this paper, it would be very interesting to see if chronic inhibition of dopaminergic neurons had a similar or different effect, as too little dopaminergic activity may also cause degeneration in some cases.

We agree that assessing chronic inhibition is valuable, and this is an important area for future research.

**Reviewer #3 (Recommendations For The Authors):**
All the mice used in the study are not listed in the methods section. For example, the GCaMP6f floxed mice discussed in the results section are not listed in the methods. Also, the breeding scheme used for the different mouse lines needs to be described. For example, did the DAT-Cre mice carry one or two alleles?

Both the DAT^IRES^Cre and GCaMP6f floxed (Ai148) Jax mouse line numbers and RRIDs are included in the methods. DAT^IRES^Cre mice carried two alleles.

In the methods section, the amount of virus injected needs to be mentioned.

This information has been added to the methods section.

In all result graphs, please include the individual data points so that the readers can see the distribution of the data and quickly see the sample size.

Graphs have been updated to include all individual data points. For line graphs, the distribution is communicated by the error bars, while the n is in the legends.

The authors provide running wheel data in supplementary figure 1A to validate that chemogenetic activation of dopamine neurons leads to increased locomotor activity. The results shown in the figure appear to be qualitative as no average data is presented. The authors should provide average data from all mice tested.

Average IP response data for all mice assessed for running wheel activity has been included in Figure S1.